# Sec24 phosphorylation regulates autophagosome abundance during nutrient deprivation

**Saralin Davis[1], Juan Wang[1], Ming Zhu[1], Kyle Stahmer[2], Ramya Lakshminarayan[2], Majid Ghassemian[3,4], Yu Jiang[5], Elizabeth A Miller[2,6], Susan Ferro-Novick[1]***

[1]Department of Cellular and Molecular Medicine, University of California, San Diego, San Diego, United States; [2]Department of Biological Sciences, Columbia University, New York, United States; [3]Department of Chemistry and Biochemistry, University of California, San Diego, San Diego, United States; [4]Biomolecular and Proteomics Mass Spectrometry Facility, University of California, San Diego, San Diego, United States; [5]Department of Pharmacology and Chemical Biology, University of Pittsburgh School of Medicine, Pittsburgh, United States; [6]MRC Laboratory of Molecular Biology, Cambridge, United Kingdom

**Abstract** Endoplasmic Reticulum (ER)-derived COPII coated vesicles constitutively transport secretory cargo to the Golgi. However, during starvation-induced stress, COPII vesicles have been implicated as a membrane source for autophagosomes, distinct organelles that engulf cellular components for degradation by macroautophagy (hereafter called autophagy). How cells regulate core trafficking machinery to fulfill dramatically different cellular roles in response to environmental cues is unknown. Here we show that phosphorylation of conserved amino acids on the membrane-distal surface of the *Saccharomyces cerevisiae* COPII cargo adaptor, Sec24, reprograms COPII vesicles for autophagy. We also show casein kinase 1 (Hrr25) is a key kinase that phosphorylates this regulatory surface. During autophagy, Sec24 phosphorylation regulates autophagosome number and its interaction with the C-terminus of Atg9, a component of the autophagy machinery required for autophagosome initiation. We propose that the acute need to produce autophagosomes during starvation drives the interaction of Sec24 with Atg9 to increase autophagosome abundance.

*For correspondence: sfnovick@ucsd.edu

**Competing interests:** The authors declare that no competing interests exist.

## Introduction

Autophagy is a highly conserved catabolic process that uses membrane traffic to target proteins and organelles for degradation. Basal levels of autophagy continuously replenish the cellular pool of amino acids and other metabolites to maintain homeostasis. However, when cells are starved for nutrients, autophagy is quickly upregulated. This upregulation leads to a dramatic intracellular reorganization to meet the high demand for membrane required to form autophagosomes, distinct organelles that target cellular components for degradation (*Nakatogawa et al., 2009*). Induction of autophagy leads to the formation of a double-membrane structure, called the isolation membrane, that forms adjacent to a pre-autophagosome structure (PAS) where autophagy related proteins (Atg) are recruited in a hierarchical manner (*Nakatogawa et al., 2009*). As the isolation membrane expands, it engulfs cytoplasmic proteins and organelles targeted for degradation before it seals to form an autophagosome. The autophagosome then fuses with the vacuole/lysosome, releasing its contents for degradation (*Lamb et al., 2013*; *Nakatogawa et al., 2009*). Although the assembly

**eLife digest** When cells experience stressful conditions, such as a shortage of nutrients, they can digest their own material via a 'self-eating' process called autophagy and then recycle the products for further use. When autophagy is triggered, a new membrane structure called the autophagosome forms within the cell as it engulfs the material that is to be digested. The autophagosome delivers these materials to a compartment where they are broken down into smaller parts and the resulting raw materials are reused as needed.

The membranes that make up the autophagosome are derived from other membranes within the cell. These include small membrane-bound compartments called vesicles, which carry proteins from one part of the cell to another, or to the outside of the cell. COPII vesicles, for example, carry out the first transport step in the pathway that leads out of the cell – the so-called secretory pathway. Recently it was found that, when cells are starving, COPII vesicles can be diverted to the autophagy pathway and provide a source of membrane to build the autophagosome. However, it was not understood how the membrane of a COPII vesicle is reprogrammed so that it can interact with the cellular machinery that builds autophagosomes.

Using genetic and biochemical methods, Davis et al. have now teased apart the distinct roles of COPII vesicles in autophagy and the secretory pathway in budding yeast. The results show that a protein called Sec24, a component of the coat on the vesicles, interacts with another protein called Atg9, which is needed for the first steps of autophagosome formation. Davis et al. observed that Sec24 could be modified by the attachment of phosphate groups at a distinct site on the surface of Sec24. This modification promotes Sec24 to interact with Atg9 and increases the number of autophagosomes that form when cells are starving. Davis et al. also found that the enzyme casein kinase 1 is one of the enzymes responsible for attaching phosphate groups to Sec24.

Following on from this work, it will be important to test whether modification of vesicle coat proteins is a widespread mechanism for reprogramming membranes for different uses in other situations as well.

pathway of the Atg proteins is known, the mechanism by which membranes are directed to the autophagy pathway remains a central unanswered question in the field.

Autophagosome biogenesis has been linked to COPII vesicles and an ER subdomain called the ER exit sites (ERES) where COPII vesicles are formed (*Graef et al., 2013*; *Suzuki et al., 2013*; *Tan et al., 2013*; *Ge et al., 2014*; *Wang et al., 2015*; *Lemus et al., 2016*). How COPII vesicles, which are faithfully targeted to the Golgi, can be reprogrammed to function on an alternate trafficking pathway during nutrient deprivation remains enigmatic. COPII coated vesicle formation is initiated at the ER with the recruitment of an inner coat layer comprising the Sec23/Sec24 complex. Coat polymerization and vesicle budding occur when Sec23/Sec24 recruits a second complex (Sec13/Sec31) that forms the outer shell of the coat. Sec24, the major cargo adaptor of the COPII coat, recruits biosynthetic cargo and SNAREs (which mediate vesicle fusion) into vesicles that are delivered to the Golgi (*Lord et al., 2013*). After vesicle fission, the coat lingers on the vesicle to facilitate vesicle targeting to the Golgi (*Cai et al., 2007*; *Lord et al., 2011*). COPII vesicle budding mutants, as well as other mutants that disrupt ER-Golgi traffic, also disrupt autophagy (*Hamasaki et al., 2003*). However, a direct functional link between COPII vesicles, the COPII coat and autophagy has been difficult to demonstrate.

Multiple COPII coat subunits, including Sec23 and Sec24, are phosphorylated by Hrr25 (*Bhandari et al., 2013*; *Lord et al., 2011*), a kinase required for ER-Golgi traffic and autophagy (*Lord et al., 2011*; *Murakami et al., 1999*; *Wang et al., 2015*; *Yu and Roth, 2002*). Previously, we showed coat phosphorylation is required for COPII vesicle fusion (*Lord et al., 2011*; *Wang et al., 2015*). Given the recently identified role of COPII vesicles in autophagosome formation, and the observation that Hrr25 is required for autophagy, we asked if coat phosphorylation also functions to regulate vesicle traffic during autophagy. Here we find that phosphorylation of the membrane distal surface of Sec24 promotes the interaction of Sec24 with the C-terminus of Atg9, which is needed for autophagy. Failure to phosphorylate this Sec24 site leads to a decrease in autophagosome number,

but not autophagosome expansion. This phosphorylation event is independent of the assembly of the Atg machinery at the PAS. Together these studies reveal a surprising role for coat phosphorylation in reprogramming core trafficking machinery to fulfill a separate function during starvation induced stress.

## Results

### Identification of Sec24 phosphorylation sites

To address whether coat phosphorylation allows COPII vesicles to function in autophagy versus ER-Golgi traffic, we purified the COPII inner coat from yeast cells induced for autophagy. Our analysis initially focused on Sec24 as it is the major COPII cargo adaptor. Using mass spectrometry, we identified 27 high confidence Sec24 phosphorylation sites in vivo (*Supplementary file 1*) and subsequently tested if they specifically affect autophagy but not ER-Golgi traffic. Many of the identified Sec24 phosphosites were conserved in the closely related paralog Iss1, which is also a cargo adaptor (*Kurihara et al., 2000*) (*Supplementary file 1*).

Two of the conserved Sec24 phosphosites (S730 and S735) map to a region of the protein that comprises one of the four well-characterized yeast cargo-binding sites (*Miller et al., 2003*, *2005*; *Pagant et al., 2015*). The so-called A-site (*Figure 1—figure supplement 1A*) packages a SNARE, Sed5, needed for ER-Golgi traffic and autophagy (*Miller et al., 2005*; *Mossessova et al., 2003*; *Tan et al., 2013*), marking these residues as candidate regulatory sites. COPII vesicles formed in vitro with Sec24-S730A/S735A showed normal capture of Sed5 and other cargo, whereas those formed with Sec24-S730D/S735D contained reduced amounts of Sed5 and were modestly impaired in their fusion efficiency with the Golgi (*Figure 1—figure supplement 1B,C*). Additionally, a strain harboring the Sec24-S730D/S735D mutations had reduced autophagic activity, whereas the Sec24-S730A/S735A mutation had no effect (*Figure 1—figure supplement 1D,E*). Because the Sec24-S730D/S735D mutations had effects on both ER-Golgi traffic and autophagy, these sites are unlikely to specify the pathway the vesicle takes and instead highlight the importance of Sed5 packaging into COPII vesicles in both pathways. A second set of conserved phosphosites, S645/S678, located near a known cargo binding site in mammalian Sec24 (mSec24) (*Figure 1A*), did not disrupt the packaging of any cargo tested (*Figure 1—figure supplement 2*).

### A novel role for Sec24 during autophagy

Having ruled out known cargo-packaging sites on Sec24 as a functional switch for COPII vesicles from ER-Golgi traffic to autophagy, we next screened a panel of alanine mutations in the remaining phosphorylation sites conserved in Iss1. The effects of mutations in these sites on autophagy were monitored in a *sec24Δiss1Δ* double mutant background by assaying Pho8Δ60 activity. Pho8Δ60 is a cytosolic form of vacuolar alkaline phosphatase that is delivered to and activated in the vacuole when autophagy is induced (*Klionsky, 2007*). Only combinations of T324A, T325A and T328A were defective in autophagy (*Figure 1B*, red box). These three sites form a patch located on the membrane distal surface of Sec24 (*Figure 1A*), making them unlikely to regulate cargo packaging. None of the additional 13 phosphosites tested showed autophagy defects (*Figure 1B*).

When we dissected the role of individual residues on this membrane distal surface of Sec24, single and double mutant combinations of T324A/T325A/T328A showed a range of defects in Pho8Δ60 activity, with the triple alanine mutant showing the most dramatic defect (*Figure 2A*). We were unable to test the full range of phosphomimetic combinations, as some caused lethality (*Figure 1C* and *Supplementary file 1*). However, none of the viable phosphomimetic mutants showed a defect in autophagy (*Figure 2B,C*). Sec24-T324A/T325A/T328A (hereafter referred to as Sec24-3A) was further characterized because it had the most dramatic defect in Pho8Δ60 activity. To confirm that phosphorylation of the Sec24 membrane distal surface is required for autophagy, we examined a second autophagy marker, Atg8. Cytosolic GFP-Atg8 is lipidated and incorporated into membrane at the PAS before being delivered to the vacuole (*Klionsky et al., 2007*). Translocation of GFP-Atg8 to the vacuole, following nitrogen starvation, was significantly reduced in a strain expressing Sec24-3A (*Figure 2D*). This defect was confirmed using western blot analysis by monitoring the cleavage of GFP-Atg8 to GFP (*Figure 2E*). The GFP-Atg8 translocation defect was less severe in the presence of Iss1 (*Figure 2D*, right), suggesting functional complementation by the paralogous protein.

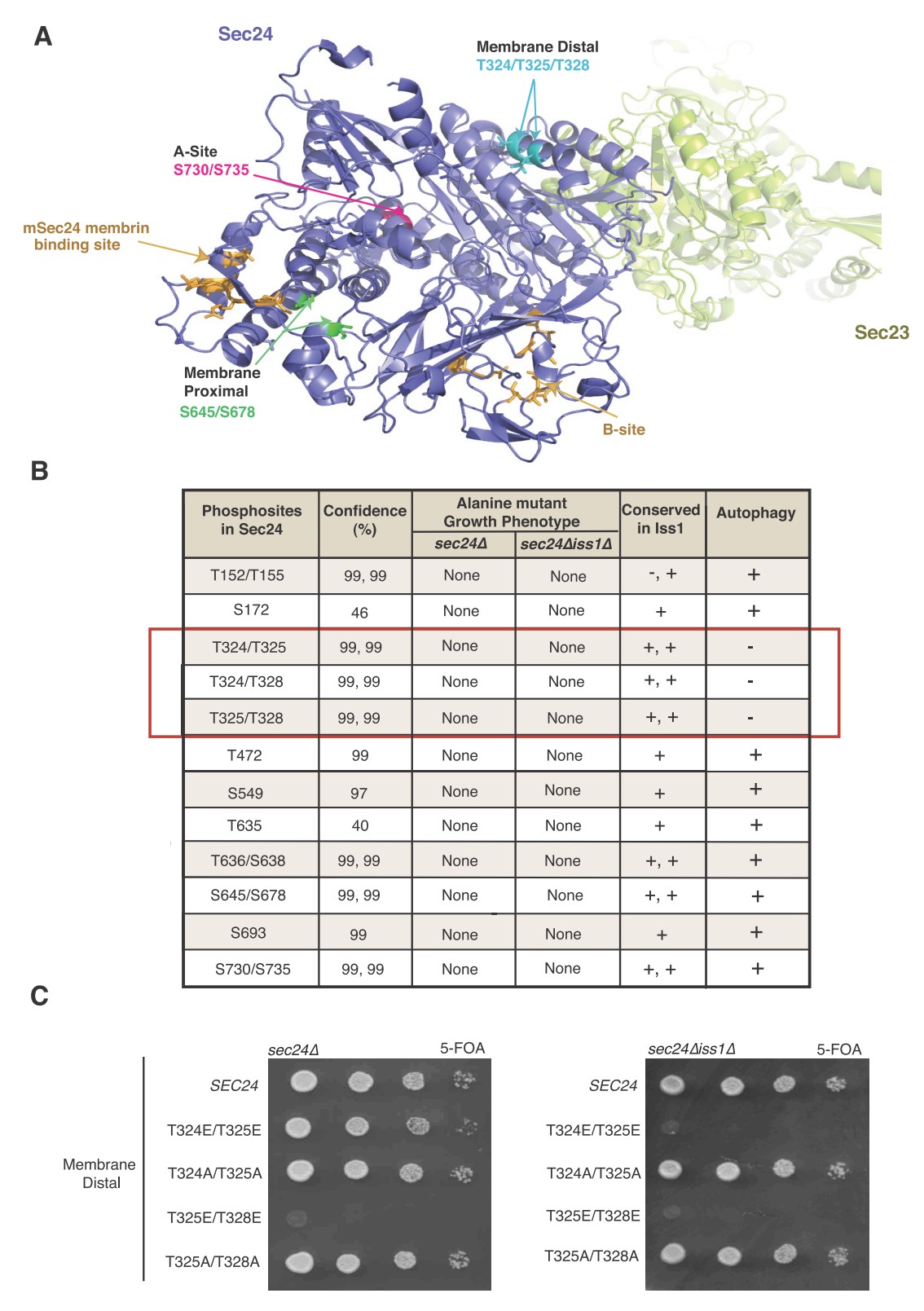

**Figure 1.** Identification of Sec24 phosphorylation sites required for autophagy. (**A**) Ribbon diagram of Sec23 (lime) and Sec24 (lavender) with groups of key Sec24 phosphorylation sites (green, pink, and teal). The mSec24 membrin-binding site and conserved cargo binding B-site are colored light brown. (**B**) Sec24 phosphorylated residues conserved in Iss1 were mutated to alanine and introduced into *sec24Δ* and *sec24Δiss1Δ* deletion strains as described in the Materials and Methods. Sec24 and Iss1 were aligned using MAFFT alignment program and residues were considered conserved if either serine

*Figure 1 continued on next page*

*Figure 1 continued*

or threonine. Mutants in the *sec24Δiss1Δ* background were screened for autophagy defects 2 to 4 hr after nitrogen starvation using vacuolar alkaline phosphatase activity as a marker. Assays were performed in triplicate biological replicates and mutants were considered defective if p<0.05 using Student's paired t-test. (C) Plasmids encoding *SEC24* (WT) or mutant *sec24* were expressed in SFNY2201 (left) or SFNY2202 (right) and grown on 5-FOA at 25°C to select against the WT balancing plasmid as described in the Materials and methods.

The following figure supplements are available for figure 1:

**Figure supplement 1.** Phosphorylation of Sec24 S730/S735 disrupts Sed5 packaging.

**Figure supplement 2.** Sec24 S645/S678 does not affect cargo packaging.

We next demonstrated that phosphorylation of the membrane distal sites on Sec24 is specifically required for autophagy by examining ER-Golgi traffic in cells containing Sec24-3A. The processing of carboxypeptidase Y (CPY), as it traffics from the ER (p1), Golgi (p2), and vacuole (m) is kinetically indistinguishable from wild-type (*Figure 2F*), demonstrating that phosphorylation of the Sec24 membrane distal patch regulates a novel function of Sec24 that is specific to autophagy. Supporting the model that a subpopulation of COPII vesicles is diverted from the secretory pathway during autophagy, ER to Golgi traffic was delayed, but not blocked, during starvation (*Figure 2—figure supplement 1A*). Expression of Sec24-3A did not rescue this delay even in an *iss1Δ* strain background (*Figure 2—figure supplement 1B,C*) suggesting that additional sites and/or factors also participate in reprogramming COPII vesicles during starvation.

Since Sec24 robustly co-purifies with Sec23, our mass spectrometry data also included information on Sec23, which contained three phosphosites on an alpha helix structurally equivalent to the Sec24 membrane distal patch (*Figure 2—figure supplement 2A*). Mutation of these Sec23 sites (T146/S147/S149) caused no defects in growth (*Figure 2—figure supplement 2B*), Pho8Δ60 activity (*Figure 2—figure supplement 2C*) or translocation of GFP-Atg8 to the vacuole (*Figure 2—figure supplement 2D*), suggesting the effects we observed on autophagy are specific to Sec24 and its paralog, Iss1.

## Phosphorylation of T324/T325/T328 regulates autophagosome number during starvation

We next wanted to address whether phosphorylation of the Sec24 membrane distal sites is specifically required for autophagosome formation during starvation-induced upregulation of autophagy. To begin to address this question, structured illumination microscopy (SIM) was used to examine GFP-Atg8 puncta formation in the absence (nutrient rich) or presence of rapamycin. GFP-Atg8 puncta mark autophagosomes, as well as the PAS and isolation membrane (*Kirisako et al., 1999*). In nutrient-rich media, autophagosome-like structures, called Cvt vesicles, form. These vesicles traffic a precursor form of aminopeptidase I (prApe1) to the vacuole in an Atg11-dependent manner (*He et al., 2006*), where it is proteolytically activated (mApe1). GFP-Atg8 puncta were smaller (*Figure 3A*), less numerous, and dependent on Atg11 (*Figure 3B*) in nutrient rich medium. Consistent with the possibility that the GFP-Atg8 puncta may represent Cvt vesicles, Atg11 was only required for their formation in rich medium and not in rapamycin-treated cells (compare *Figure 3B, C*).

Cells expressing Sec24-3A were not defective in GFP-Atg8 puncta formation in nutrient rich medium (*Figure 3B*), however, fewer puncta formed in the rapamycin-treated Sec24-3A cells (*Figure 3C*). While Sec24-3A significantly reduced the number of puncta formed, it did not significantly affect their size (*Figure 3D*). Consistent with Sec24-3A not affecting Cvt vesicle formation, processing of Ape1 was unaffected in *sec24* alanine mutants (*Figure 3—figure supplement 1A*). Similar results were obtained when we blocked COPII vesicle budding at 37°C in the temperature-sensitive mutant *sec12-4* mutant (*Figure 3—figure supplement 1B,C*). The observation that COPII vesicles are not needed on the Cvt pathway is consistent with our earlier studies (*Wang et al., 2015*) and those of *Ishihara et al. (2001)*

We used a second assay to address the effect of Sec24-3A on the frequency of autophagosome formation during starvation. Autophagosome number and size can be assessed using transmission

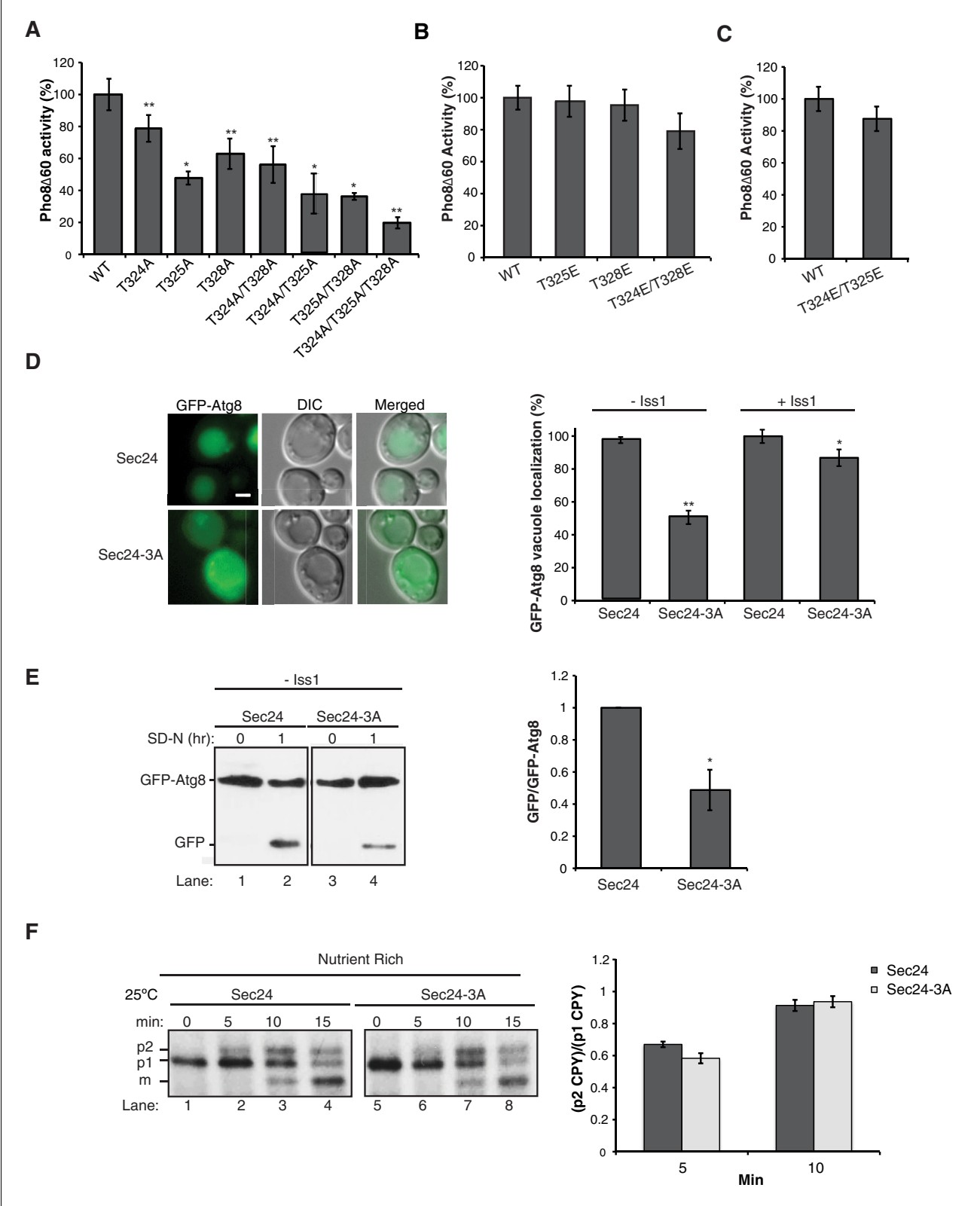

**Figure 2.** Phosphorylation of T324/T325/T328 in Sec24 is required for autophagy, but not ER-Golgi transport. (**A**) Vacuolar alkaline phosphatase activity was assayed in lysates prepared from a *sec24Δiss1Δ* deletion strain harboring *sec24* alanine mutations. The activity of wild-type (WT) 2 hr after starvation was set as 100% and 0 time-point values were subtracted. Averages and s.e.m. are shown for 3 (or four for T324A/T325A) biological replicates. p-values=0.006 (T324A), 0.012 (T325A), 0.009 (T328), 0.008 (T324A/T328A), 0.02 (T324A/T325A), 0.01 (T325A/T328A), 0.006 (T324A/T325A/T328A),

*Figure 2 continued on next page*

*Figure 2 continued*

Student's paired t-test. (B, C) As in (A) except activity was assayed in extracts from phosphomimetic mutations in *sec24Δiss1Δ* (B) or *sec24Δ* (C) deletion strains. Averages and s.e.m. are shown for three biological replicates. p-values=0.88 (T325E), 0.78 (T328E), 0.32 (T324E/T328E), 0.26 (T324E/T325E), Student's paired t-test. (D) The translocation of GFP-Atg8 to the vacuole was examined 1 hr after nitrogen starvation at 25°C in *sec24Δiss1Δ* and *sec24Δ* deletion strains in either the presence of WT Sec24 or Sec24-3A. Representative images (left) and quantification from 300 cells (right) are shown. Scale bar 2 μm. WT was set to 100% for each experiment and had an average vacuolar localization of 76% (*sec24Δiss1Δ*) and 86% (*sec24Δ*). Averages and s. e.m. are shown for three biological replicates. p-values=0.006 (*sec24Δiss1Δ*), 0.02 (*sec24Δ*), Student's paired t-test. (E) Cleavage of GFP-Atg8 was examined in *sec24Δiss1Δ* cells expressing Sec24 or Sec24-3A 1 hr after starvation at 25°C (left). The ratio of free GFP to GFP-Atg8 was quantitated. The cleavage in WT was set to 1 (right). Averages and s.e.m. are shown for three biological replicates. p-value = 0.015, Student's unpaired t-test. (F) *sec24Δiss1Δ* cells expressing Sec24 (lanes 1–4) or Sec24-3A (lanes 5–8) were pulse-labeled for 4 min and chased for the indicated times (left). The p1 (ER), p2 (Golgi) and m (vacuolar) forms of CPY are labeled. Quantitation of the ratio of p2/p1 CPY for the 5 and 10 min time points are shown (right). Averages and s.e.m. are shown for three biological replicates. p-values=0.08 (5 min), 0.66 (10 min), Student's unpaired t-test. *p<0.05; **p<0.01.

The following figure supplements are available for figure 2:

**Figure supplement 1.** ER-Golgi transport is delayed during autophagy.

**Figure supplement 2.** Phosphorylation of the Sec23 membrane distal sites is not required for autophagy.

electron microscopy by analyzing autophagic bodies, fully formed autophagosomes that have fused with the vacuole. Upon deletion of the *PEP4* gene, which encodes a protease that is required for the activation of multiple vacuolar hydrolases, autophagic bodies accumulate in the vacuole (*Backues et al., 2014*).

After starvation, fewer autophagic bodies accumulated in cells expressing Sec24-3A compared to WT Sec24 (*Figure 3E,F*). Although autophagic body number was significantly reduced, autophagic body size was not affected (*Figure 3E,F,G*). Together these findings indicate that phosphorylation of the Sec24 membrane distal sites regulates autophagosome number, while autophagosome size or expansion is largely unaffected. The role of Sec24 in autophagy is likely to be conserved in mammalian cells as T324 and T328 are conserved in mSec24A (*Figure 4A*).

## Phosphorylation regulates the interaction of Sec24 with the C-terminus of Atg9

We previously proposed that COPII vesicles fuse with Atg9 vesicles at the PAS (*Tan et al., 2013*). When autophagy is induced, Atg9 vesicles traffic from the late Golgi to the PAS and localize adjacent to the ER exit sites that produce COPII vesicles (*Graef et al., 2013*; *Suzuki et al., 2013*; *Yamamoto et al., 2012*). Interestingly, similar to our observations with Sec24, Atg9 was recently shown to regulate autophagosome number, but not size (*Jin et al., 2014*). Proteomics also revealed an interaction between multiple COPII coat subunits and Atg9 in detergent lysates (*Graef et al., 2013*). However, it remained unclear from these studies which coat subunit mediated this interaction and whether it was required for autophagy. Therefore, we first confirmed that Sec24 co-precipitates with Atg9 and tested whether this interaction is regulated by starvation. Sec24 was immunoprecipitated from cells expressing Atg9-13myc that were grown in nutrient rich conditions (SMD) or starved for nitrogen (SD-N), and the precipitate was then blotted with anti-myc antibody. Atg9-13myc co-immunoprecipitated with Sec24, but not the pre-immune control (*Figure 4B*). Moreover, approximately 2.5-fold more Atg9-13myc co-immunoprecipitated with Sec24 from lysates prepared from nitrogen starved cells (*Figure 4B*), demonstrating that this interaction is upregulated during starvation.

Atg9 is a six transmembrane protein with N-terminal, C-terminal and core or middle (M) cytoplasmic domains (*Figure 4C*). While the middle and C-terminal domains are present in mammalian cells, the N-terminus is largely absent (*Young et al., 2006*). In order to determine which domain of Atg9 interacts with Sec24, the cytoplasmic domains of Atg9 were fused to GST and the purified fusion proteins were incubated in vitro with purified Sec23/Sec24 complex. The Sec23/Sec24 complex was used for these studies, since Sec24-His$_6$ is unstable in the absence of Sec23. Sec24 was predominantly unphosphorylated, as it was purified without phosphatase inhibitors and stored in the freezer following purification. A GST fusion to a Sec31 fragment (aa878-1114), which was previously

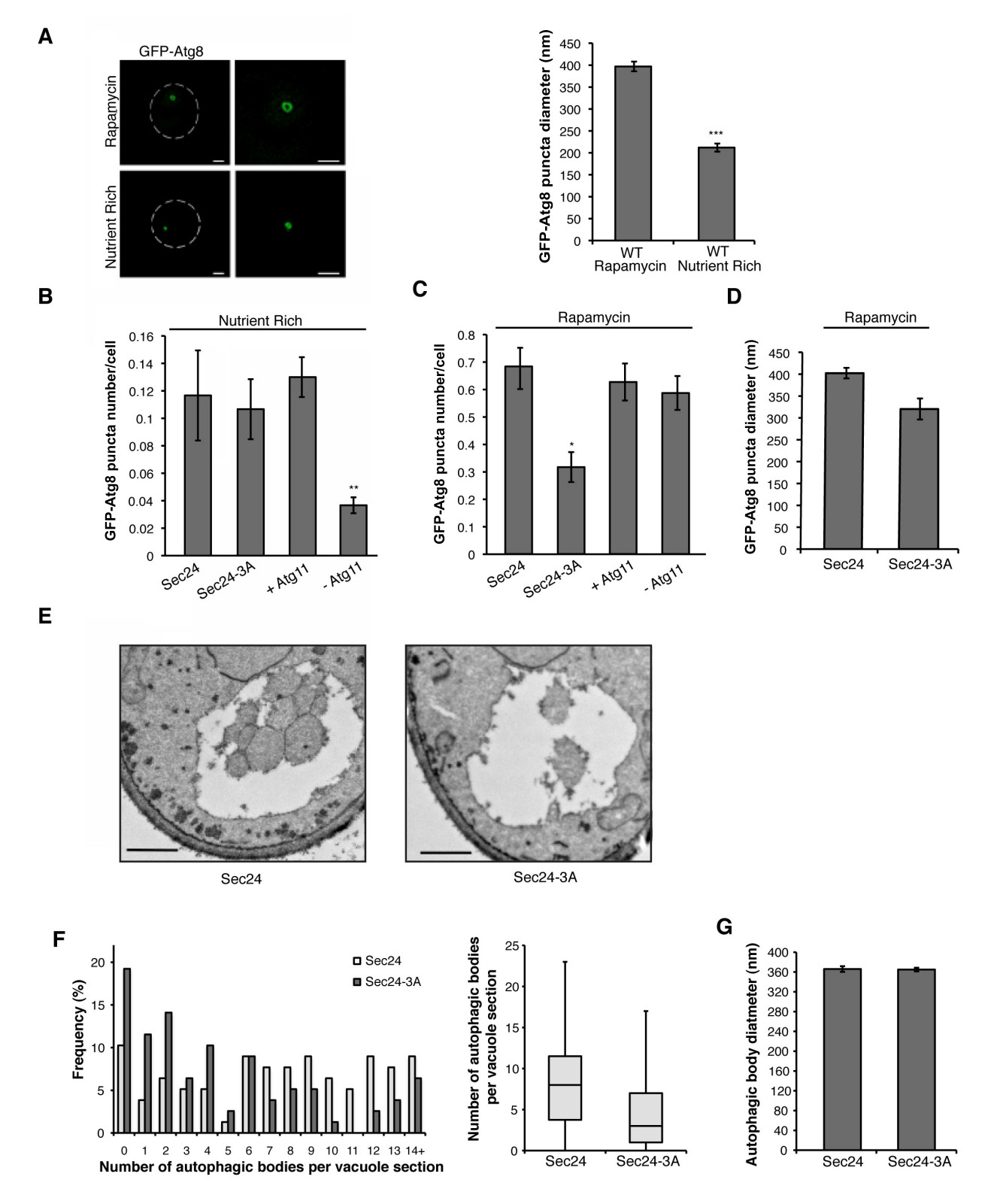

**Figure 3.** Phosphorylation of Sec24 regulates autophagosome frequency during starvation. (**A**) Representative images from WT cells expressing GFP-Atg8 treated with 400 ng/ml rapamycin for 1 hr at 25°C (left top) or untreated (left bottom). Deconvolved images are shown. Scale bar, 1 μm. 100 GFP-Atg8 puncta were measured in WT cells treated with or without rapamycin (right). Averages and s.e.m. are shown for three biological replicates; p=0.0002, Student's unpaired t-test. (**B**) WT Sec24 and Sec24-3A expressed in the *sec24Δiss1Δ* deletion strain, and WT (+Atg11) and *atg11Δ* cells (-
*Figure 3 continued on next page*

*Figure 3 continued*

Atg11) expressing GFP-Atg8 were imaged and the number of puncta per cell was quantitated from 300 cells. Averages and s.e.m. are shown for three biological replicates; p-values=0.81 (Sec24-3A), 0.009 (-Atg11), Student's unpaired t-test. (C) As in (B) except cells were treated with 400 ng/ml rapamycin for 1 hr at 25°C. Averages and s.e.m. are shown for three biological replicates; p-values=0.02 (Sec24-3A), 0.68 (-Atg11), Student's unpaired t-test. (D) As in (C) only the diameter of 100 GFP-Atg8 puncta was measured from cells expressing WT Sec24 and Sec24-3A in *sec24Δiss1Δ* deletion strains treated with 400 ng/ml rapamycin for 1 hr at 25°C. Averages and s.e.m. are shown for four biological replicates; p-value = 0.057, Student's unpaired t-test. (E) Representative images of autophagic bodies in cells expressing Sec24 (left) and Sec24-3A (right) in *sec24Δiss1Δpep4Δ* deletion strains after 1.5 hr of nitrogen starvation at 30°C. Scale bar represents 500 nm. (F) Histogram showing the distribution of the number of autophagic bodies per vacuole section in Sec24 and Sec24-3A. The number of autophagic bodies was quantitated for 78 vacuole sections for each strain (left). p-value = 0.00012; Mann-Whitney Test. Box plot of the number of autophagic bodies per vacuole section. Bars show data between the lower and upper quartiles, the median is a horizontal line within the box. Whiskers indicate the smallest and largest observations (right). (G) The diameter of autophagic bodies was determined. For Sec24 N = 398, for Sec24-3A N = 342. Averages with error bars as s.e.m. are shown. *p<0.05; **p<0.01; ***p<0.001.

The following figure supplement is available for figure 3:

**Figure supplement 1.** GFP-Atg8 puncta formation is affected in the *sec12-4* mutant during autophagy induction.

shown to interact with the Sec23/Sec24 complex (*Bi et al., 2007*), served as a positive control. The C-terminus of Atg9 bound to Sec23/Sec24, while the N-terminal and middle hydrophilic Atg9 domains did not (*Figure 4D*, *Figure 4—figure supplement 1A*). Furthermore, binding of Sec23/Sec24 to the C-terminus increased with increasing concentrations of the Sec23/Sec24 complex and appeared to be saturable (*Figure 4E*, *Figure 4—figure supplement 1B*). This interaction was also dependent on Sec24, as Sec23 alone did not interact with GST-Atg9C (*Figure 4F*, *Figure 4—figure supplement 1C*). As the crystal structure of Sec23 and Sec24 are identical whether they are in a complex or not, allosteric effects are unlikely (*Bi et al., 2002*).

Next we used the phosphomimetic mutations to ask if the interaction between Sec24 and Atg9 is enhanced by phosphorylation of the membrane distal sites. While the most dramatic effects on autophagy were seen with the *sec24* triple alanine mutant (*Figure 2*), the *sec24* triple phosphomimetic mutant is inviable. Therefore, to cover all three phosphosites of interest in our binding studies, we used Sec23/Sec24-T325E and Sec23/Sec24-T324E/T328E. Consistent with the notion that phosphorylation enhances the Sec24-Atg9 interaction, both T325E and T324E/T328E increased the interaction of Sec24 with GST-Atg9C (*Figure 5A*, *Figure 5—figure supplement 1A*).

To determine whether the Sec24-Atg9 interaction is regulated by phosphorylation in vivo, wild-type or mutant *sec24* cells co-expressing Atg9-13myc were starved for nitrogen and Sec24 was immunoprecipitated in the presence of phosphatase inhibitors. Sec24-T324A/T325A (*Figure 5B*), but not Sec24-T324E/T325E (*Figure 5—figure supplement 1B*) disrupted the interaction of Sec24 with Atg9-13myc. This defect was more pronounced in Sec24-3A, where Sec24-T328 is also mutated (*Figure 5C*). Sec24-3A did not impair the trafficking of Atg9 to the PAS (*Figure 5—figure supplement 2A,C*), indicating that Sec24 does not regulate Atg9 traffic. Together these findings imply that the C-terminus of Atg9 interacts with phosphorylated Sec24 via its membrane distal surface after Atg9 has been recruited to the PAS.

Microscopy-based studies have linked ERES to autophagosome formation (*Graef et al., 2013*; *Suzuki et al., 2013*), however, the ERES are also a subdomain of the ER where COPII vesicles bud (*Budnik and Stephens, 2009*). To ask if Sec24-3A disrupts the formation of ERES, the localization of Sec13-GFP was examined in cells expressing wild-type Sec24 or Sec24-3A. Sec13-GFP, a component of the outer COPII coat, predominantly localizes to ERES (*Shindiapina and Barlowe, 2010*). ERES localization of Sec13-GFP was not affected by Sec24-3A in nutrient rich or starvation conditions, suggesting Sec24-3A does not disrupt ERES formation (*Figure 5—figure supplement 3*). Thus, our findings imply that the major function of the Sec24 membrane distal surface is to regulate the interaction of COPII vesicles with the Atg machinery.

## Hrr25 regulates autophagy via Sec24

Hrr25, the only kinase known to phosphorylate Sec24 in yeast, is required for COPII vesicle fusion in ER-Golgi traffic and autophagy (*Lord et al., 2011*; *Wang et al., 2015*). Recent studies, however,

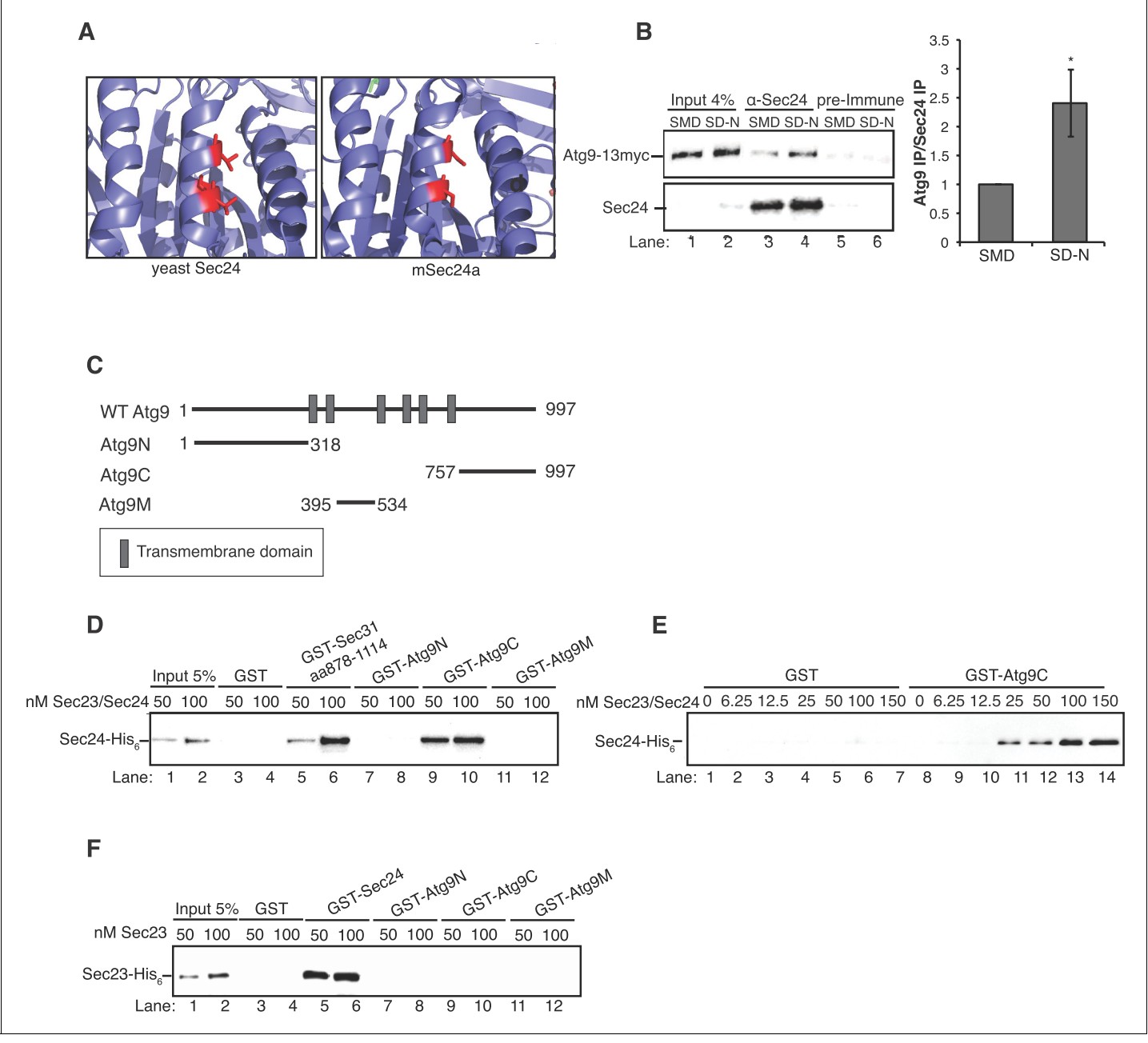

**Figure 4.** The Sec23/Sec24 complex binds the C-terminus of Atg9. (**A**) Structure of yeast Sec24 and mSec24a with conserved residues in membrane distal sites (red). (**B**) *ypt7Δ* cells expressing Atg9-13myc were grown in nutrient rich (SMD) or starvation (SD-N) media for 4 hr and Sec24 was immunoprecipitated and blotted for Atg9-13myc (left). Precipitated Atg9-13myc was quantitated and normalized to the amount of Sec24 in the precipitate. SMD was set as one for each experiment. *ypt7Δ* cells were used as autophagosomes fail to fuse with the vacuole in the absence of Ypt7 and accumulate in the depleted cells (*Kirisako et al., 1999*). Averages and s.e.m. are shown for four biological replicates, p-value = 0.037, Student's unpaired t-test. (**C**) Schematic showing cytosolic domains of Atg9. (**D**) Equimolar amounts (200 nM) of purified GST, GST-Sec31 (aa878-1114) or GST-Atg9 fragments were incubated with 50 or 100 nM of Sec23/Sec24-His$_6$. (**E**) Equimolar amounts (100 nM) of purified GST or GST-Atg9C was incubated with increasing amounts of Sec23/Sec24-His$_6$. (**F**) Same as (**D**) except His$_6$-Sec23 was used.

The following figure supplement is available for figure 4:

**Figure supplement 1.** Ponceau staining of in vitro bindings in *Figure 4D, E and F*.

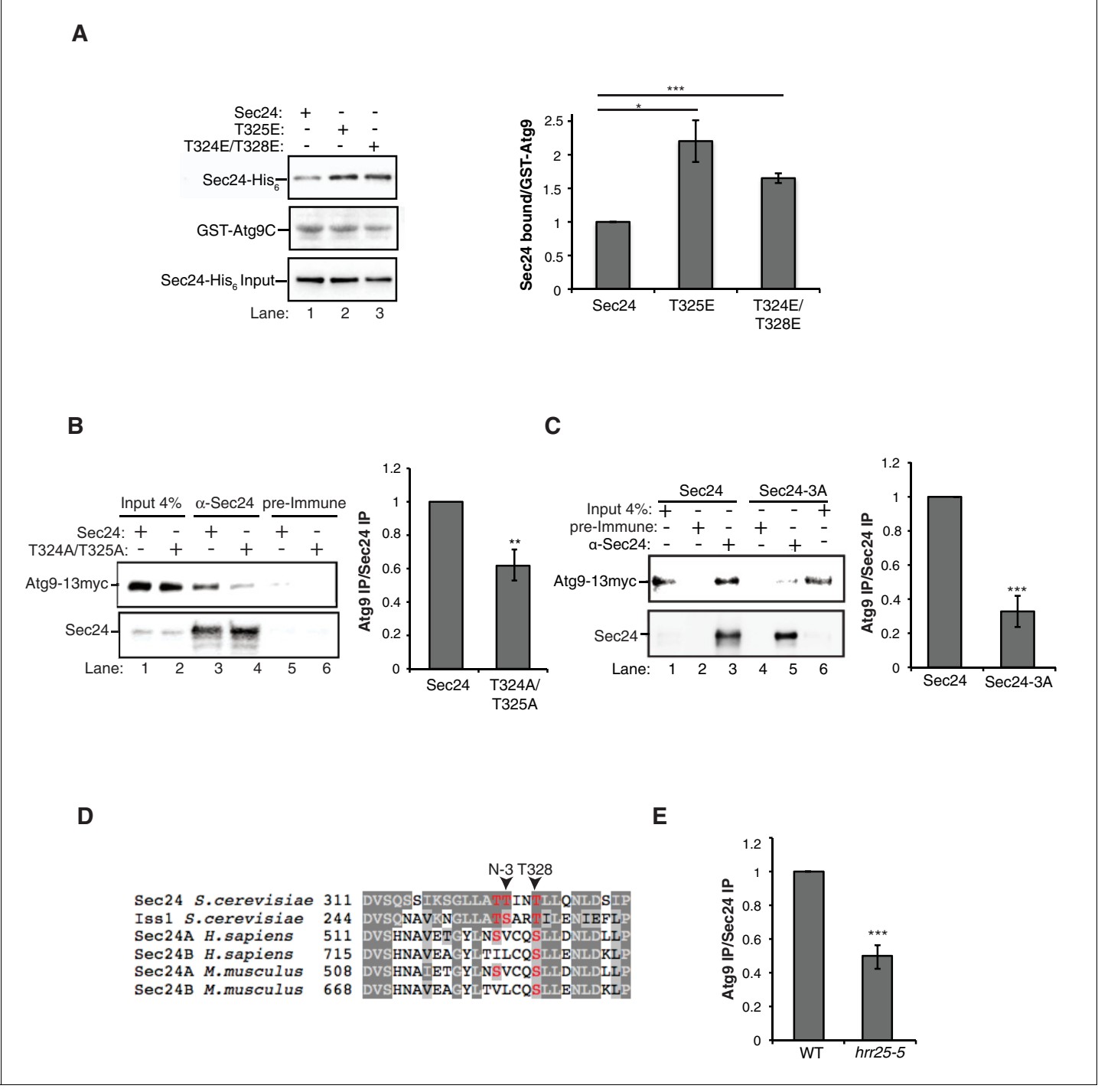

**Figure 5.** Phosphorylation of the Sec24 membrane distal sites regulates the Sec24-Atg9 interaction. (**A**) GST-Atg9C (200 nM) was incubated with 37.5 nM of WT Sec23/Sec24-His₆, Sec23/Sec24-T325E-His₆ or Sec23/Sec24-T324E/T328E-His₆ (left). Ratio of Sec24 bound to GST-Atg9C was quantified from three biological replicates. Averages and s.e.m. are shown (right). WT Sec23/Sec24 was set as one for each experiment; p-value = 0.018 (T325E), 0.0008 (T324E/T328E), Student's unpaired t-test. (**B**) Sec24 (WT) and Sec24-T324A/T325A or (**C**) Sec24-T324A/T325A/T328A (Sec24-3A) were immunoprecipitated from lysates expressing Atg9-13myc as described in the Materials and Methods. Precipitated Atg9-13myc was quantitated and normalized to the amount of Sec24 in the precipitate. WT Sec24 was set as one for each experiment. Averages and s.e.m. are shown for 4 (**B**) or 5 (**C**) biological replicates; p-value = 0.006 (**B**), 0.0008 (**C**) Student's unpaired t-test. (**D**) Alignment of the region surrounding T324/T325/T328 (shown in red) with Sec24 orthologues. (**E**) Same as (**B**) except Sec24 was immunoprecipitated from WT or *hrr25-5* lysates. Averages and s.e.m. are shown for five biological replicates. p-value = 0.0002, Student's unpaired t-test. **p<0.01; ***p<0.01; ***p<0.001.

*Figure 5 continued on next page*

*Figure 5 continued*

The following figure supplements are available for figure 5:

**Figure supplement 1.** GST negative control for in vitro binding in **Figure 5A**.
**Figure supplement 2.** Sec24-3A does not affect Atg assembly at the PAS.
**Figure supplement 3.** Sec24-3A does not affect ERES formation.
**Figure supplement 4.** Representative blots for quantitation shown in **Figure 5E**.
**Figure supplement 5.** Autophagic body number is reduced in the *hrr25-5* mutant.

have suggested Hrr25 has an additional role in autophagy, upstream of COPII vesicle fusion at the PAS (*Wang et al., 2015*). Specifically, we found that while COPII vesicles accumulate at the PAS in some mutants that disrupt autophagy, they failed to accumulate in the *hrr25-5* mutant. Additionally, epistasis studies revealed Hrr25 acts upstream of the key autophagy kinase, Atg1 (*Wang et al., 2015*).

Interestingly, Sec24-T328 fits the CK1 consensus motif (pS/pT-X-X-S/T, *Figure 5D*) (*Knippschild et al., 2005*) and was found with low confidence to be phosphorylated by Hrr25 in vitro. Consistent with a role for Hrr25 in phosphorylating the Sec24 membrane distal patch, less Atg9 co-immunoprecipitated with Sec24 from an *hrr25-5* mutant lysate (*Figure 5E*, *Figure 5—figure supplement 4*). Additionally, like the *sec24* triple mutant (*Figure 3E,F,G*), we observed a reduction in autophagic body number, but not size in the *hrr25-5* mutant using transmission electron micros-copy (*Figure 5—figure supplement 5*). To determine if Hrr25 regulates the Sec24-Atg9 interaction via Sec24 phosphorylation, we asked whether the Sec24 phosphomimetic mutations could rescue the Sec24-Atg9 interaction defect in *hrr25-5*. Sec24 T325/T328 was chosen for this analysis as it con-tains T328, which is phosphorylated by Hrr25. WT Sec24, Sec24 T325A/T328A or Sec24 T325E/T328E was ectopically expressed in *hrr25-5* cells containing Atg9-13myc and Sec24 was immunopre-cipitated. Sec24 T325E/T328E significantly rescued the Sec24-Atg9 interaction in *hrr25-5*, whereas Sec24 T325A/T328A did not (*Figure 6A*). Additionally, Sec24 T325E/T328E alleviated the autophagy defect in *hrr25-5* as the vacuolar localization of GFP-Atg8 induced by starvation was partially rescued by Sec24 T325E/T328E, but not Sec24 T325A/T328A (*Figure 6B*). To confirm the fluorescence results, cleavage of GFP-Atg8 was also examined. Sec24 T325E/T328E almost fully rescued the GFP cleavage defect in *hrr25-5*, while Sec24 T325A/T328A had no effect (*Figure 6C*, *Figure 6—figure supplement 1*). Therefore, although Hrr25 is required for COPII vesicle fusion (*Lord et al., 2011*; *Wang et al., 2015*), phosphorylation of Sec24 is a primary function of this kinase during starvation induced autophagy. These findings also indicate the Sec24-Atg9 interaction is needed for autophagy.

Despite the role of Hrr25 in regulating the Atg9-Sec24 interaction, neither Hrr25 expression, nor activity was altered upon nitrogen starvation (*Figure 6—figure supplement 2*). However, as CK1 kinases act preferentially on substrates that are already phosphorylated at the N-3 position (*Figure 5D*) (*Knippschild et al., 2005*), during autophagy Hrr25 may work with other kinases that create additional Sec24 phosphosites. To determine if Hrr25 mediated phosphorylation of Sec24 is required for Atg complex assembly (*Figure 5—figure supplement 2B*), we visualized the PAS recruitment of a member of each of the major groups of Atg proteins required for autophagosome formation in the *hrr25-5* and *sec24* triple alanine mutants. We found no effect of *hrr25-5* (*Figure 6—figure supplement 3*) or Sec24-3A (*Figure 5—figure supplement 2A,C*) on the PAS recruitment of Atg2, Atg5, Atg9, Atg13 or Atg14. Consistent with our findings, blocking COPII vesicle formation in the *sec12-4* mutant also had no effect on the PAS localization of Atg2 and Atg14 (*Figure 6—figure supplement 4*). Thus, while COPII vesicles are needed for autophagy, they are dispensable for the assembly of the Atg hierarchy. Together these findings demonstrate that Sec24 phosphorylation does not indirectly affect autophagy by disrupting the trafficking of Atg proteins to the PAS.

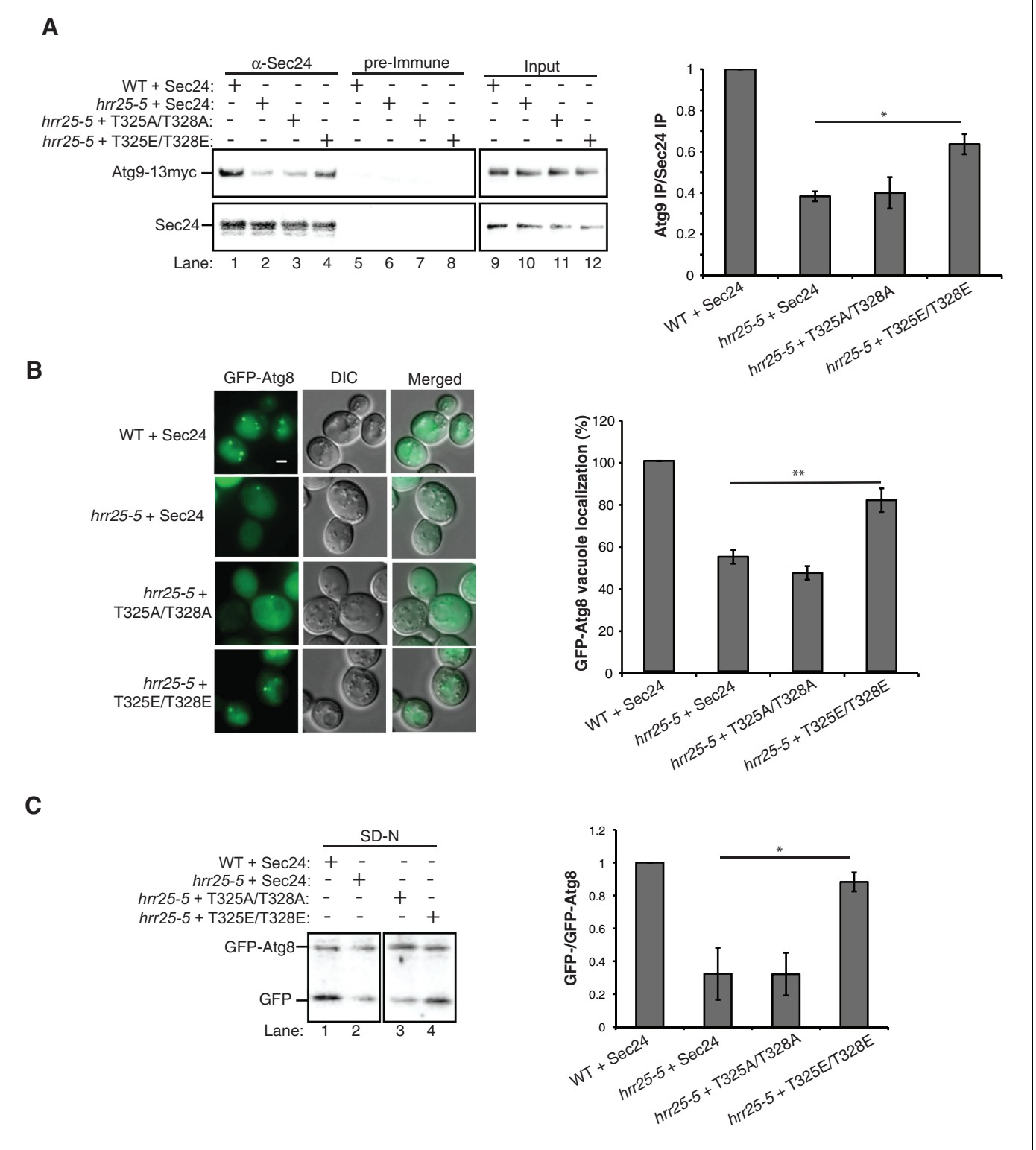

**Figure 6.** Hrr25 regulates autophagy via phosphorylation of the Sec24 membrane distal sites. (**A**) Sec24 was immunoprecipitated from WT or *hrr25-5* cells expressing Atg9-13myc and either WT Sec24 or Sec24 T325A/T328A or Sec24 T325E/T328E. Precipitated Atg9-13myc was quantitated and normalized to the amount of Sec24 in the precipitate. WT was set as one for each experiment. Averages and s.e.m. are shown for three biological replicates. p-value = 0.01 (*hrr25-5* + T325E/T328E), 0.81 (*hrr25-5* + T325A/T328A), Student's unpaired t-test. (**B**) Vacuolar localization of GFP-Atg8 was examined 2 hr after nitrogen starvation at 37°C in WT or *hrr25-5* cells expressing either WT Sec24 or Sec24 T325A/T328A or Sec24 T325E/T328E. Scale bar, 2 μm (left). Over 300 cells were quantitated from three biological replicates. WT was set as 100% for each experiment. Averages and s.e.m. are
*Figure 6 continued on next page*

*Figure 6 continued*

shown. p-value = 0.006 (*hrr25-5* + T325E/T328E), 0.17 (*hrr25-5* + T325A/T328A), Student's unpaired t-test. (**C**) Cleavage of GFP-Atg8 in *hrr25-5* cells expressing WT Sec24 or Sec24 T325A/T328A or Sec24 T325E/T328E were examined 2 hr after nitrogen starvation at 37°C (left). The ratio of GFP to GFP-Atg8 was quantitated from three biological replicates. The cleavage in WT was set to 1. Averages and s.e.m. are shown. p-value = 0.03 (*hrr25-5* + T325E/T328E), 0.99 (*hrr25-5* + T325A/T328A), Student's unpaired t-test. *p<0.05; **p<0.01.

The following figure supplements are available for figure 6:

**Figure supplement 1.** Nutrient rich controls for GFP-Atg8 cleavage in *Figure 6C*.

**Figure supplement 2.** Hrr25 is not regulated during starvation.

**Figure supplement 3.** The *hrr25-5* mutant does not affect Atg assembly at the PAS.

**Figure supplement 4.** The *sec12-4* mutant does not affect Atg2 or Atg14 PAS localization.

## Discussion

Although significant progress has been made in defining the upstream events leading to the assembly of Atg proteins at the PAS (*Nakatogawa et al., 2009*), the membrane rearrangements that occur during autophagy remain poorly understood. Here we show that phosphorylation of a conserved regulatory domain of the major COPII cargo adaptor Sec24 reprograms the function of COPII vesicles by modulating its interaction with the C-terminus of Atg9, a key component of the autophagy machinery. The C-terminus of Atg9 is present in vertebrates and is essential for autophagy (*He et al., 2008*; *Young et al., 2006*). Atg9 functions early in autophagosome initiation (*Suzuki et al., 2007*; *Yamamoto et al., 2012*) and affects autophagosome number, but not size (*Jin et al., 2014*). Similarly, failure to phosphorylate the Sec24 regulatory sites, which disrupts the Sec24-Atg9 interaction, specifically affects autophagosome number. Consistent with our proposal that autophagosome formation requires the fusion of COPII vesicles with Atg9 vesicles (*Tan et al., 2013*; *Figure 7*), we see an accumulation of COPII coated structures at the PAS in the *atg9Δ* mutant (*Figure 7—figure supplement 1*).

For technical reasons, we have been unable to directly address if the Sec24 membrane distal sites are phosphorylated as a consequence of inducing autophagy. However, given that these sites are only required for autophagosome formation during starvation-induced autophagy, and both phosphorylation and autophagy enhance the Sec24-Atg9 interaction, it seems likely the membrane distal Sec24 surface is specifically phosphorylated when autophagy is induced. As autophagosomes form in proximity to each other in nutrient rich and starved cells, the regulated phosphorylation of Sec24 would prevent the inappropriate fusion of COPII vesicles with Atg9 membranes during constitutive growth.

Phosphorylation of the Sec24 membrane distal surface is regulated at least in part by Hrr25. We have shown Hrr25 is required for autophagy and functions upstream of COPII vesicle delivery to the PAS (*Wang et al., 2015*), but how Hrr25 regulates COPII vesicles on this pathway has been unclear. The findings we present here show that, during autophagy, Hrr25 acts through Sec24 by regulating its interaction with Atg9. As Hrr25 has been linked to the TOR network (*Breitkreutz et al., 2010*), a key negative regulator of autophagy (*Loewith and Hall, 2011*), it is tempting to speculate that Hrr25 works with one or more kinases that act downstream of Tor during nutrient deprivation. We propose that the heightened need to rapidly produce autophagosomes during starvation leads to phosphorylation of the Sec24 membrane distal surface, which enhances the Sec24-Atg9 interaction, resulting in an increase in autophagy to maintain homeostasis (*Figure 7*). These events are independent of the assembly of the Atg hierarchy that occurs when autophagy is induced.

Previous work has established that ERES and COPII vesicles are required for autophagy in both yeast and mammalian cells (*Ge et al., 2014*; *Graef et al., 2013*; *Lemus et al., 2016*; *Suzuki et al., 2013*; *Tan et al., 2013*; *Wang et al., 2015*). However, because it has been difficult to fully tease apart the function of COPII vesicles on the secretory pathway from their role in autophagy, their contribution to autophagy has been problematic to address. Our finding that phosphorylation of a novel

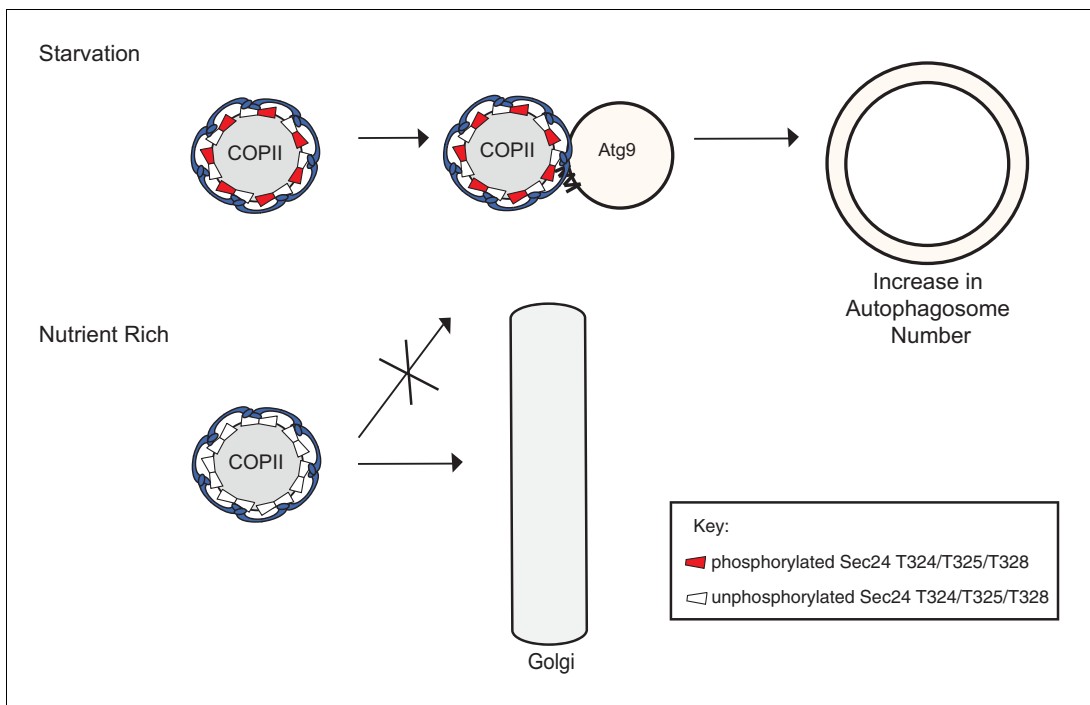

**Figure 7.** Phosphorylation of a conserved regulatory surface of Sec24 enhances the ability of the COPII coat to recognize the autophagy machinery. Phosphorylation of the Sec24 membrane distal sites regulates the interaction of Sec24 with the C-terminus of Atg9. During starvation, the Sec24-Atg9 interaction is needed to increase autophagosome number. If Sec24 is not phosphorylated, it is unable to efficiently interact with Atg9 and COPII vesicles traffic to the Golgi.

The following figure supplement is available for figure 7:

**Figure supplement 1.** Sec13-GFP accumulates at the PAS in the *atg9Δ* mutant.

conserved regulatory surface of Sec24 is specifically required for autophagy and not ER-Golgi transport conclusively demonstrates that COPII vesicles play a direct role in autophagy, rather than an indirect role in maintaining the trafficking of autophagy machinery through the secretory pathway. Additionally, we can now ascribe an autophagy-specific role to COPII vesicles in enhancing autophagosome number during nutrient deprivation, which is likely to be conserved.

In mammalian cells, nucleation of autophagosomes occurs in the vicinity of the ER and the Atg9 compartment (*Karanasios et al., 2016*). Interestingly, membrane fractionation identified the ER-Golgi intermediate compartment (ERGIC), which resides adjacent to the ER, as the site of lipidation of the Atg8 homologue LC3 (*Ge et al., 2014*). During starvation, the ERES were found to translocate to the ERGIC fraction, and although the ERGIC is typically a location of COPI vesicle budding (*Lorente-Rodríguez and Barlowe, 2011*), COPII vesicle budding from this ERGIC fraction was shown to be required for LC3 lipidation (*Ge et al., 2014*). These observations suggested that, in higher eukaryotes, the COPII vesicles used in autophagy are spatially separated from those that traffic to the Golgi. The findings we report here could explain how these spatially segregated COPII vesicles engage the Atg machinery.

Other components of the secretory pathway have also been implicated in autophagy (*Amaya et al., 2015*; *Geng et al., 2010*; *Nair et al., 2011*). It remains to be seen whether these components directly contribute membrane for autophagosome formation or if they only affect the trafficking of autophagy machinery, such as Atg9, to sites of autophagosome formation. It is known that another component of the autophagy machinery, Atg8, controls autophagosome size, but not number (*Xie et al., 2008*). However, it is unclear if Atg8 is sufficient to drive autophagosome expansion or if additional membranes are required for this event.

In conclusion, our findings highlight an unexpected role for phosphorylation in regulating the reorganization of membrane trafficking pathways during starvation and demonstrate that the COPII coat is a key target of this regulation. The findings we present here ascribe a new role for Sec24 and the COPII coat, and provide a possible explanation for why the coat is retained on the vesicle subsequent to vesicle scission (*Cai et al., 2007*; *Lord et al., 2011*). Identification of a cellular mechanism that redirects the flow of membrane during autophagy makes it possible to now study how these complex membrane rearrangement events culminate in the formation of a distinct organelle, the autophagosome. Future work will be needed to determine if the diverted COPII vesicles contain autophagy-specific cargo, or if the role of these vesicles in autophagy is solely to bring certain lipids and SNAREs to this pathway.

## Materials and methods

### Media, growth conditions and yeast strains

Strains used in this study are listed in *Supplementary file 2*. Yeast cells were grown at 25°C in rich media (YPD: 1% yeast extract, 2% peptone, and 2% dextrose) or synthetic minimal media (SMD: 0.67% yeast nitrogen base, 2% dextrose and auxotrophic amino acids as needed). Nitrogen starvation was induced in synthetic minimal medium lacking nitrogen (SD-N: 0.17% yeast nitrogen base without amino acids, 2% dextrose). For galactose induction of Sec24, the growth conditions are described in the next section. For solid media, agar was added to a final concentration of 2%.

### Mass spectrometry

Sec24-His$_6$ was purified from SFNY2181 (*Supplementary file 2*) as described previously (*Kurihara et al., 2000*) with the following modifications. Cells were grown overnight to log phase in SC-Ura-Leu medium with 10% glycerol as the carbon source, and induced at a starting OD$_{600}$ of 0.6–0.8 with 0.2% galactose for 5 hr at 30°C. These growth conditions were found to induce autophagy in the absence of rapamycin. Cells were lysed in approximately 20 ml HSLB (0.75 M potassium acetate, 50 mM HEPES pH 7.0, 0.1 mM EGTA, 20% glycerol) with protease and phosphatase inhibitor cocktails (Sigma, St. Louis, MO). The cleared lysate was incubated with Ni-NTA beads for 1 hr at 4°C and washed with 20 ml B-II and 20 ml B-III (*Kurihara et al., 2000*). Sample preparation for mass-spectrometry was carried out as described before (*Guttman et al., 2009*), then liquid chromatography coupled tandem mass spectrometry analysis (LC-MS/MS) was performed as described previously (*Meyer et al., 2014*). The high confidence Sec24 mass spectrometry data from three runs are compiled in *Supplementary file 3*. Note, potential sites buried in the Sec24 structure were not analyzed further.

### Generation of sec24 mutants

Mutations in pSFN1915 (*SEC24, HIS3, CEN*) were made using the QuikChange Site-directed mutagenesis kit (Agilent technologies, Santa Clara, CA) and all mutations were confirmed by DNA sequencing. Plasmids were then introduced into SFNY2201 and SFNY2202 (*Supplementary file 2*) and grown for two rounds on 5-fluoroorotic acid (5-FOA) plates at 25°C to select against *pLM22* (*SEC24, URA3, CEN*). To observe growth defects, *sec24* mutants were compared to WT *SEC24* at 25°C after two rounds of 5-FOA. For purification of mutant coat proteins, the mutations were made on *pSFNB1895* (*GAL1-SEC24-His, LEU2, CEN*) and co-transformed with *pSFNB1894* (*GAL1-SEC23, URA3, CEN*) into SFNY2367 (*Supplementary file 2*).

### Analysis of phosphosites on the Sec23/Sec24 structure

All structures were accessed through the protein data bank (PDB) and analyzed with PyMol software. The Sec23/Sec24 structure, PDB ID 1M2V, was reported in a previous study (*Bi et al., 2002*). The Sec24 structure with the Sed5 peptide, PDB ID 1PD0, was reported in a previous study (*Mossessova et al., 2003*).

### In vitro vesicle budding and fusion assays

COPII proteins (Sar1, Sec23/Sec24 and Sec13/Sec31) were purified (*Miller et al., 2003*) and used to generate vesicles from microsomal membranes prepared as described (*Barlowe et al., 1994*).

Vesicle budding assays were performed as described previously (*Miller et al., 2002*). Vesicle fusion was monitored by measuring $\alpha-1,6$-mannose modification of $^{35}S$-labeled pro-$\alpha$-factor as described previously (*Barlowe et al., 1994*).

## Pho8Δ60 assay

Alkaline phosphatase assays were performed as previously described (*Klionsky, 2007*). Cells were grown overnight at 25°C to log phase, $OD_{600}$ between 0.7 and 1.0, washed with 10 ml SD-N medium and incubated in SD-N medium for 2 to 4 hr at 25°C or 37°C to induce autophagy. 2.5 $OD_{600}$ units of cells were collected and washed, and lysed in 250 µl of lysis buffer (20 mM PIPES pH 7.2, 0.5% TritonX-100, 50 mM KCl, 100 mM potassium acetate, 10 mM MgS0$_4$, 10 µM ZnSO$_4$, and 1 mM PMSF) using glass beads. Lysates, with a protein concentration around 0.5 mg/ml, were spun for 5 min at 13,000 rpm, and 100 µl of lysate was assayed at 37°C in 400 µl reaction buffer (1.25 mM p-nitrophenyl phosphate, 250 mM Tris-HCl pH 8.5, 0.4% Triton X-100, 10 mM MgSO$_4$, and 10 µM ZnSO$_4$). The reaction was stopped with 500 µl of stop buffer (1 M glycine/KOH pH 11.0), and the $OD_{400}$ value was determined. The data were normalized to protein concentration using the Bradford method and IgG as a standard.

## Fluorescence microscopy

For GFP-Atg8 vacuolar localization, cells were grown overnight at 25°C in SC-Ura to early log phase, $OD_{600}$ between 0.6 and 1.0. Cells were washed and resuspended in SD-N and incubated for 1 hr at 25°C (Sec24-3A), 30 min at 37°C (Sec24-S730/S735), or 2 hr at 37°C (*hrr25-5* mutant). For co-localization of Ape1-RFP with Atg proteins, cells were grown overnight to early log phase at $OD_{600}$0.6–1.0 and treated with 400 ng/mL rapamycin for 1 hr at 25°C (Sec24-3A), or starved for nitrogen for 2 hr at 37°C (*hrr25-5* and *sec12-4* mutants). Cells were then visualized at 25°C with a Zeiss Axio Imager Z1 fluorescence microscope using a 100×1.3 NA oil-immersion objective. Images were captured with a Zeiss AxioCam MRm digital camera and analyzed with AxioVision software.

To examine GFP-Atg8 puncta by structured illumination (SIM) microscopy, cells were grown overnight at 25°C in SC-Ura to early log phase, $OD_{600}$ between 0.6 and 1.0. For autophagy induction, cells were treated with 400 ng/ml rapamycin for 1 hr at 25°C or for 1 hr at 37°C for temperature-sensitive mutants. Cells were pelleted and incubated in 3.7% formaldehyde for 30 min at 25°C and visualized on an Applied Precision DeltaVision OMX Super Resolution System using an Evolve 512 EMCCD camera. The data was acquired and processed using Delta Vision OMX Master Control software and SoftWoRx reconstruction and analysis software. To determine autophagosome size, deconvolved images were analyzed with Image J software.

## GFP-Atg8 cleavage assay and Ape1 processing

To observe cleavage of GFP-Atg8 during autophagy, cells were grown overnight to early log phase, washed, resuspended in SD-N and incubated for 1 hr at 25°C (Sec24-3A) or 2 hr at 37°C (*hrr25-5* mutant). The cells (2.5 $OD_{600}$ units) were then pelleted, resuspended in 0.1 M NaOH and incubated for 5 min at room temperature. The samples were spun and heated in sample buffer for 5 min at 95°C before SDS PAGE. To monitor Ape1 processing, cells were grown in nutrient rich media to early log phase and lysed as described above.

## CPY pulse chase

Cells were grown overnight in minimal media at 25°C to early log phase and 16 $OD_{600}$ units of cells were pelleted and resuspended in 3.6 ml of fresh minimal media. For starved samples, cells were washed and shifted to SD-N for 1 hr at 25°C. 16 $OD_{600}$ units of cells were then pelleted and resuspended in 3.6 ml of fresh SD-N. Cells were pulse labeled with 400 µCi of S$^{35}$-methionine for 4 min at 25°C, and 700 µl of the cell suspension was removed and added to 700 µl of ice-cold 20 mM sodium fluoride/sodium azide (0 min time-point). 250 µl of chase mix (250 mM methionine, 250 mM cysteine) was added to the remaining sample, and then 700 µl of cells were removed at 5, 10 and 15 min. Cells were pelleted and washed with 1 ml of cold 10 mM sodium fluoride/sodium azide, resuspended in 150 µl spheroplasting buffer (1.4 M sorbitol, 100 mM sodium phosphate pH 7.5, 0.35% β-mercaptoethanol and 0.2 mg/ml zymolyase) and incubated at 37°C for 45 min. Spheroplasts were spun for 3 min at 6500 rpm and heated for 5 min at 95°C in 100 µl 1% SDS. 900 µl of PBS plus 2%

Triton X-100 was added to the lysates before they were spun for 15 min at 14,000 rpm. CPY antibody (3 µl of anti-Rabbit serum prepared against CPY) was added to 920 µl of cleared lysate and incubated for 1 hr at 4°C with rotation. 50 µl of 50% Protein-A sepharose was added and incubated for 1 hr. The protein-A beads were washed twice with 1 ml of PBS, followed by two washes with 1 ml of 1% β-mercaptoethanol and heated in 70 µl of 1x sample buffer for 5 min at 95°C. Samples were normalized to cpm in the cell lysate, then loaded onto an 8% SDS-PAGE gel and processed for autoradiography. Protein bands were quantified using Image J software.

## Electron microscopy

Cells were grown overnight in YPD to an $OD_{600}$ of 1.0 and shifted to SD-N for 1.5 hr at 30°C (Sec24-3A) or 1.5 hr at 37°C (*hrr25-5* mutant). 30 $OD_{600}$ units of cells were pelleted, resuspended in 1 mL of 1.5% $KMnO_4$ and incubated for 30 min at 4°C with nutation. Cells were then pelleted and resuspended in 1 mL of 1.5% $KMnO_4$ and incubated overnight at 4°C with nutation. Samples were dehydrated in ethanol, embedded in Durcupan epoxy resin (Sigma-Aldrich) and sectioned at 60 nm on a Leica UCT ultramicrotome. Sections were picked up on Formvar and carbon-coated copper grids and stained with 2% uranyl acetate for 5 min and Sato's lead stain for 1 min. Grids were viewed using a Tecnai G2 Spirit BioTWIN transmission electron microscope equipped with an Eagle 4k HS digital camera (FEI, Hilsboro, OR). Autophagic body number and size were determined with Adobe Photoshop and Image J software as described previously (*Backues et al., 2014*).

## Co-immunoprecipitation of Sec24 and Atg9

Cells were grown overnight to early log phase. For starvation, cells were shifted to SD-N for 4 hr at 30°C, or for the *hrr25-5* mutant 2 hr at 37°C. 100 $OD_{600}$ units of cells were pelleted, resuspended in 2 ml of spheroplasting buffer (1.4 M sorbitol, 100 mM sodium phosphate pH 7.5, 0.35% β-mercaptoethanol and 0.5 mg/ml zymolyase) and incubated for 30 min at 37°C. Spheroplasts were loaded on top of a 5 ml sorbitol cushion (1.7 M sorbitol, 100 mM HEPES pH 7.2) and spun for 5 min at 3000 rpm. Cells were lysed in 1 ml of lysis buffer II (20 mM Hepes pH 7.4, 150 mM NaCl, excess protease inhibitors (Roche, Switzerland and Sigma mix), phosphatase inhibitors (Sigma)) with a dounce homogenizer on ice. Cell debris was cleared by a 10 min spin at 500 xg. When cross-linking was performed, lysates were incubated on ice with 100 mM dithiobis (succinimidyl propionate) for 30 min. To quench excess crosslinker, 100 mM Tris pH 7.6 was added and incubated for 15 min on ice. Triton X-100 was added to a final concentration of 1% and incubated on ice for 30 min followed by a 15 min spin at 15,000xg. To immunoprecipitate Sec24, 2 mg of lysate was incubated with 10 µl of Sec24 antibody (rabbit polyclonal prepared against GST-Sec24) or 10 µl of pre-immune serum for 2 hr at 4°C with rotation. 50 µl of 50% protein A-sepharose was added and incubated for 45 min at 4°C with rotation. The protein-A beads were pre-incubated with 1 mg/ml BSA for 30 min at 4°C before they were added to the sample to reduce background. The beads were then washed five times with 1 ml of lysis buffer with 1% Triton X-100 and heated in 40 µl of 3x sample buffer for 5 min at 95°C. Note that similar amounts of Atg9-13myc co-immunoprecipitated with Sec24 without the use of crosslinker.

## Purification of fusion proteins from bacteria for in vitro binding studies

To induce expression of GST fusion proteins, bacterial cells were incubated overnight at 18°C with 0.5 mM isopropyl β-D-1-thiogalactopyranoside. Cells were collected, resuspended in 1x phosphate-buffered saline (PBS) with 1 mM DTT and protease inhibitors, then sonicated for a total of 2 min with 15 s on/off bursts on ice. Triton X-100 was added to a final concentration of 1% and lysates were incubated on ice for 15 min. The lysates were cleared by a 15 min centrifugation at 15,000 rpm. The supernatant was incubated with 1 mL of 50% glutathione sepharose beads (GE Healthcare, United Kingdom) that was prewashed with PBS for 1 hr at 4°C with rotation before the beads were washed extensively with PBS and stored at 4°C.

His$_6$-Sec23 was purified as described above for the GST fusion proteins, except cells were lysed in 50 mM Hepes pH 7.2, 150 mM NaCl, 15 mM Imidazole, 1 mM DTT with protease inhibitors and incubated with $Ni^{2+}$-NTA resin (Qiagen, Germany). Protein was eluted from the resin with 50 mM Hepes pH 7.2, 150 mM NaCl, 250 mM Imidazole.

## In vitro bindings with GST-Atg9

Equimolar amounts (0.2 μM) of GST fusion proteins were incubated with rotation in binding buffer (50 mM HEPES pH 7.2, 150 mM NaCl, 1% Triton X-100, 1 mM $MgCl_2$, 1 mM EDTA, 1 mM DTT, protease inhibitors) for 4 hr at 4°C with increasing amounts of $His_6$-Sec23 that was purified from bacteria or Sec23/Sec24-$His_6$ purified from yeast as described before (*Miller et al., 2002*). The beads were washed 3–4 times with binding buffer and eluted in 50 μL of sample buffer by heating for 5 min at 95°C.

## Kinase assay

Cells expressing Hrr25-HA were grown overnight in SC-Ura to early log phase, $OD_{600}$0.6–0.8, then washed and shifted to SD-N medium for 1 hr at 25°C to induce autophagy. The cells were harvested by centrifugation, washed with 20 mM Tris pH 7.4, resuspended in 5 ml of spheroplasting buffer and incubated at 37°C for 30 min. Spheroplasts were pelleted through a 10 ml sorbitol cushion, resuspended in 5 ml of lysis buffer III (50 mM Tris-HCl pH 7.4, 100 mM NaCl, 5 mM EDTA, 1 mM PMSF, 1% Triton X-100, 1X protease inhibitor mixture (Roche)) and lysed with a dounce homogenizer on ice. Lysates were then cleared by a 15 min centrifugation at 14,000 rpm. To immunoprecipitate Hrr25-HA, lysates were incubated with 20 μl anti-HA resin (Sigma) for 2 hr at 4°C with rotation. The beads were washed three times with lysis buffer and two times with kinase buffer (50 mM HEPES pH 7.4, 5 mM $MgCl_2$, 0.2% NP-40 and 1 mM DTT). The kinase activity of immunopurified Hrr25-HA was assayed in a 50 μl reaction volume using 1 μg of myelin basic protein (MBP) as substrate as described before (*Wang et al., 2013*).

## Acknowledgements

We thank Deepali Bhandari for her participation in the mass spectrometry analysis of Sec24 in the initial stages of the project, Wenyun Zhou for technical assistance in the construction of *sec24* mutants, Chris Lord for guidance with the in vitro transport studies, Jodi Nunnari for the pFA6a-3xyEGFP-CaURA3 construct; Claudine Kraft for the plasmid used to express GST-Atg9C, and Seth Field for comments during the preparation of the manuscript. We also thank Ying Jones in the Electron Microscopy facility in the Department of Cellular and Molecular Medicine at UCSD, headed by Dr. Marilyn Farquhar, for EM sample preparation, and Dan Klionsky for protocols on the preparation and analysis of the EM data. We acknowledge the UCSD School of Medicine microscopy core for use of the Applied Precision Delta Vision OMX Super Resolution System. This work was supported by NIGMS and NCI under award numbers R01GM114111 and R01GM115422 to SF-N, R01GM085089 and MC_UP_1201/10 to EAM, and R21CA169186 to Y Jiang.

## Additional information

### Funding

| Funder | Grant reference number | Author |
| --- | --- | --- |
| National Institute of General Medical Sciences | GM114111 | Saralin Davis<br>Juan Wang<br>Susan Ferro-Novick |
| National Cancer Institute | CA169186 | Yu Jiang |
| National Institute of General Medical Sciences | GM085089 | Kyle Stahmer<br>Ramya Lakshminarayan<br>Elizabeth A Miller |
| National Institute of General Medical Sciences | GM115422 | Saralin Davis<br>Juan Wang<br>Susan Ferro-Novick |
| Medical Research Council | MC_UP_1201/10 | Elizabeth A Miller |

The funders had no role in study design, data collection and interpretation, or the decision to submit the work for publication.

## Author contributions
SD, Conception and design, Acquisition of data, Analysis and interpretation of data, Drafting or revising the article; JW, MZ, KS, RL, MG, Acquisition of data, Analysis and interpretation of data; YJ, Analysis and interpretation of data, Drafting or revising the article; EAM, Acquisition of data, Analysis and interpretation of data, Drafting or revising the article; SF-N, Conception and design, Analysis and interpretation of data, Drafting or revising the article

## Author ORCIDs
Saralin Davis, http://orcid.org/0000-0002-9834-8683
Susan Ferro-Novick, http://orcid.org/0000-0001-8714-7352

## Additional files

### Supplementary files
• Supplementary file 1. Identification of Sec24 phosphorylation sites. Sec24-His$_6$ was purified from yeast and subjected to tandem mass spectrometry as described in the Materials and Methods. Phosphorylation sites with a confidence level of 40% or greater were further analyzed. Phosphorylated residues were mutated to either aspartic or glutamic acid and introduced into *sec24Δ* and *sec24Δiss1Δ* deletion strains as described in the Materials and Methods. Sec24 and Iss1 were aligned using MAFFT alignment program and residues were considered conserved if either serine or threonine.

• Supplementary file 2. Key yeast strains used in this study.

• Supplementary file 3. High confidence Sec24 phosphorylated peptides. Compilation of mass spectrometry data of Sec24 phosphorylated peptides summarized in *Figure 1B* and *Supplementary file 1*.

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
