## [Decision Letter]

[Editors’ note: a previous version of this study was rejected after peer review, but the authors submitted for reconsideration. The first decision letter after peer review is shown below.]

Thank you for choosing to send your work, "Sec24 phosphorylation regulates autophagosome abundance during nutrient deprivation", for consideration at *eLife*. Your initial submission has been assessed by Randy Schekman in consultation with three reviewers. Although the work is of interest, we regret to inform you that the findings at this stage are too preliminary for further consideration at *eLife*.

The reviews were mixed but the general opinion is that the work is interesting but not yet complete. Given the nature of the additional work described in the reviews below, the feeling among the referees is that more than two months of additional experimental effort would be required to satisfy the concerns and thus we must decline to publish this work now but certainly encourage you to return a new version when you have completed the additional work. We will endeavor to seek the views of the same reviewers on such a resubmission.

Please note that we aim to publish articles with a single round of revision that would typically be accomplished within two months. This means that work that has potential, but in our judgement would need extensive additional work, will not be considered for in-depth review. We do not intend any criticism of the quality of the data or the rigor of the science. We wish you good luck with your work and we hope you will consider *eLife* for future submissions.

*Reviewer #1*

The first part that describe the identification of Sec24 phosphorylation sites (T324/T325/T328) specifically important for autophagy is clear. However, the second part proposing that Hrr25-mediated phosphorylation of Sec24 reprograms COPII vesicles to function in autophagy is too preliminary. As the interaction of COPII subunits and Atg9 is not novel, the data presented here represent a limited advance and rather preliminary to support the model shown in Figure 6 (Figure 1 is not immediately relevant to this study). It is also unclear how the Sec24-Atg9 interaction regulates the number, rather than the size, of the autophagosome.

1) It is unclear how Sec24 phosphorylation regulates the COPII-Atg9 vesicle interaction as shown in the model in Figure 6. To examine the requirement of the Sec24-Atg9 binding in autophagosome biogenesis, it is important to identify the binding sites in both Sec24 and Atg9 and generate binding-deficient mutants.

2) The association of the COPII vesicles and Atg9 vesicles should be extensively investigated using biochemical methods and dependency on the Sec24 phosphorylation and the Sec24-Atg9 binding should also be tested.

3) Figure 3: Showing autophagy-dependent cleavage of GFP-Atg8 by immunoblotting will complement the results.

4) It is not clear whether Hrr25 contributes to autophagy indeed through Sec24 phosphorylation. This can be tested by expression of the phosphomimetic Sec24 SE mutant in hrr25-5 mutant cells. The binding between Sec24 SE and Atg9 should be tested in hrr25 mutant cells (Figure 5).

*Reviewer #2*

My opinion is mixed. The study provides a nice assessment of phosphosites in Sec24 and mutation of the distal threonine residues (324, 328, 328) clearly influence autophagosome abundance. However, I felt the evidence that starvation activates Hrr25 to directly phosphorylate these residues and that phosphorylated Sec24 then reprograms COPII vesicles for targeting to Atg9 containing membranes was a bit preliminary or overstated. At this stage I don't think it is up to par for *eLife* but I could have missed something.

From the consultation session:

In reading over your reviews we appear to see similar strengths but share some of the same concerns. I think a current limitation is in clearly placing Sec23-Sec24 activity in autophagosome biogenesis. Phosphorylation of Sec24 appears to play a regulatory role but this could be in somehow re-configuring ERES or in recruiting factors or directing COPII vesicles to a specific target. To devise assays/methods to address this issue is likely to take more than a couple of months which is why my enthusiasm was moderate.

*Reviewer #3*

The membrane supplying autophagosome biogenesis has been difficult to clarify. Recent evidence in yeast and mammals indicate the involvement of COPII vesicles as a contribution to the autophagic membrane. However, how COPII vesicles are diverted from ER-Golgi trafficking to support autophagosome biogenesis was not clear. Some previous evidence shed light on the possible mechanism. For example, the Ge L et al. indicated a spatial regulation by generating the COPII from the ERGIC to separate from the COPII for trafficking which is from the ERES (Ge L et al. *eLife* 2014) in mammalian cells. Yeast cells lack a special ERGIC. Nonetheless, COPII is still essential for autophagosome biogenesis as shown by the last author's work as well as the Ohsumi group. Since there is no obvious solution to spatially separate the autophagic COPII from the ER-Golgi trafficking COPII in yeast, different mechanisms should be involved. Based on the authors' previous work, they hypothesized that phosphorylation of COPII components should be a reasonable mechanism accounting for the diversion of COPII to autophagy. In the current study, the authors analyzed the phosphorylation of Sec24 because Graef et al. found an association between Sec24 and Atg9 in a previous large scale proteomic analysis (Graef et al. MBoC 2013). They identified the phosphorylation sites of Sec24 and created different loss or gain of function mutations. They found that three membrane distal phosphorylation sites are required for the number of autophagosomes formed during starvation possibly due to their association with Atg9. The Hrr25 is the kinase required for Sec24 phosphorylation and the proposed process shown in the final model. In general, the story is a nice extension of their previous studies. The new advance is to employ the phosphorylation to explain the selection of COPII function. More experiments are necessary to validate the process.

1) It is not clear at which stage does the Sec24 phosphorylated COPII act. Although the authors showed no effect on Atg9 on PAS, other marker should also be tested such as Atg14, Atg1, Atg17, Atg2, Atg18 and Atg5. Also, it is not shown if Sec24 mutations, such as the 3A or 3E that affects autophagy, also influence the localization of Sec24 on the PAS.

2) The binding between Sec24 and Atg9 is weak. And the consequence of the interaction is not clear without more biochemical data and interaction site information. The author should find the cytosolic domain of Atg9 that associates with Sec24 and validate with in vitro pull down. Based on this notion, single mutation of this Atg9 domain should also achieve a similar effect of abolishment of Sec24 phosphorylation. Or, the author could express a peptide either from Atg9 or Sec24 to compete off the interaction to further confirm.

3) Although the model of COPII-Atg9 vesicle fusion looks reasonable and has appeared many times in many of their previous papers, it is not supported by any direct evidence. in vitro fusion should be performed or the fusion should be analyzed in vivo by live cell imaging. The Sec24 mutants should be employed as negative controls.

From the consultation session:

The work is preliminary although has some interesting points. Considering their previous work about the negative effect of SEC23 phosphorylation on autophagy (Wang J et al. JCB 2005) while here it is positive effect of SEC24 phosphorylation on autophagy, the role of COPII phosphorylation and how Hrr25 regulates the specific phosphorylation of SEC23 and SEC24 on different sites and different conditions is really complicated. Although it may take more than 2 months to address these concerns, I believe the authors should be able to address the major concerns of the reviewers to improve the current work. In addition, the mechanism of how Hrr25 selects SEC23 and SEC24 as well as specific phosphorylation sites on SEC24 that specifically affect autophagy pathway and ATG9 association is very important and will increase the impact of the study.

[Editors’ note: what now follows is the decision letter after the authors submitted for further consideration.]

Thank you for submitting your article "Sec24 phosphorylation regulates autophagosome abundance during nutrient deprivation" for consideration by *eLife*. Your article has been favorably evaluated by Randy Schekman (Senior Editor) and three reviewers, one of whom is a member of our Board of Reviewing Editors. The reviewers have opted to remain anonymous.

The reviewers have discussed the reviews with one another and the Reviewing Editor has drafted this decision to help you prepare a revised submission.

Summary:

This group previously reported that Hrr25 is required for autophagy, but what this kinase phosphorylates and how it contributes to autophagosome formation have been unknown. In this study, the authors report that phosphorylation of T324, T325, and T328 of the COPII subunit Sec24 is required for autophagy. Mechanistically, phosphorylation of these residues promotes the interaction between the C-terminus of Atg9 and Sec24, which is important for the increase in the number rather than the size of autophagosomes. Based on these results, the authors propose that COPII vesicles are reprogrammed to be used for autophagosome formation by Hrr25-mediated Sec24 phosphorylation in response to nitrogen starvation.

This is a resubmitted version and the manuscript has been significantly improved. In particular, the experiments in Figure 5 showing that the interaction of phosphomimic Sec24 (T324E/T328E) with Atg9 is stronger than that of non-phosphorylated Sec24 and those in Figure 6 using T325E/T328E in hrr25-5 mutant cells have strengthened the authors' conclusion that Hrr25-mediated phosphorylation of Sec24 is important for binding with Atg9 and autophagy. However, there are several points that need to be addressed, some of which were raised by a new reviewer.

Essential revisions:

1) Figure 3. It is hard to tell that the GFP-Atg8 structures indeed represent autophagosomes rather than precursor structures (phagophores) and the PAS. The size of phagophores and the PAS are smaller than that of closed autophagosomes. It would be ideal to measure the size of GFP-Atg8 structures in ypt7Δ cells, in which closed autophagosomes should accumulate.

2) Related to the above criticism, it is also difficult to distinguish small autophagosomes from the PAS under nutrient-rich conditions. The presence of small GFP-Atg8 puncta does not necessarily mean that Cvt vesicles are normally formed because they may merely represent the PAS. Thus, the authors' proposal that COPII vesicles contribute to autophagosome formation under starvation but not nutrient-rich conditions (Cvt vesicle formation) should be carefully reevaluated. In fact, it was reported that COPII mutants are defective in the Cvt pathway (Reggiori et al. Mol Biol Cell 15, 2189, 2004). The Cvt pathway should be directly examined by seeing the maturation of Ape1 in the 3A mutant.

3) It is difficult to conclude that COPII vesicles localize to the PAS by simply observing Sec13 puncta, most of which likely represent ER exit sites. It would be difficult to distinguish the ERESs and COPII vesicles by fluorescence microscopy. Other groups concluded that the PAS forms in the vicinity of the ER exit site based on results of similar experiments.

4) The title and/or Abstract should provide a clear indication of the biological system under investigation (i.e., species name or broader taxonomic group, if appropriate). Please revise your title and/or abstract with this advice in mind.

---

## [Author Response]

[Editors’ note: the author responses to the first round of peer review follow.]

[…]

*Reviewer #1*

*The first part that describe the identification of Sec24 phosphorylation sites (T324/T325/T328) specifically important for autophagy is clear. However, the second part proposing that Hrr25-mediated phosphorylation of Sec24 reprograms COPII vesicles to function in autophagy is too preliminary. As the interaction of COPII subunits and Atg9 is not novel, the data presented here represent a limited advance and rather preliminary to support the model shown in Figure 6 (Figure 1 is not immediately relevant to this study). It is also unclear how the Sec24-Atg9 interaction regulates the number, rather than the size, of the autophagosome.*

We agree that Figure 1 is not entirely central to the final conclusions of our study and have moved these data to the supplement (see Figure 1—figure supplement 1). Speaking to the novelty of the Atg9/Sec24 interaction, we note that although an earlier proteomics study linked Atg9 to multiple COPII coat subunits, it was not shown which COPII coat subunit drives this interaction. Furthermore, and most importantly, the significance of this interaction was not explored in the earlier study. Specifically, it was not demonstrated that the COPII-Atg9 interaction is required for autophagy. We now provide additional data that supports a role for Hrr25 in modulating autophagy via the Sec24 phosphosites (i.e. the suppression of Hrr25 autophagy defects by the Sec24 phosphomimetic mutations). Thus, our data shows the Atg9-COPII interaction is mediated by Sec24, regulated by the CK1 kinase Hrr25, and used as a mechanism to commit a COPII vesicle to the macroautophagy pathway. This is a novel discovery that contributes to our understanding of how membrane traffic is regulated during autophagy.

*1) It is unclear how Sec24 phosphorylation regulates the COPII-Atg9 vesicle interaction as shown in the model in Figure 6. To examine the requirement of the Sec24-Atg9 binding in autophagosome biogenesis, it is important to identify the binding sites in both Sec24 and Atg9 and generate binding-deficient mutants.*

We have now performed in vitro binding experiments with His-tagged Sec24 and Sec23 using fragments of Atg9 fused to GST. Monomeric Sec24 is unstable in the absence of Sec23, so we used purified Sec23/Sec24 complex to demonstrate a direct interaction between Atg9 and Sec23/Sec24. Additionally, using bacterially-expressed monomeric Sec23, to rule out an interaction between Atg9 and Sec23, allows us to conclude that the Atg9-COPII coat interaction requires Sec24. We note that the crystal structure of Sec23 and Sec24 are identical whether they are in complex with each other or not, making allosteric effects entirely unlikely. We further defined the site of interaction with Atg9 by domain dissection. Atg9 is a multi-spanning transmembrane protein that contains cytoplasmic N-(Atg9N; aa 2-318) and C- (Atg9C; aa 747- 997) termini. The middle of the protein is membrane-bound with a small domain (Atg9M; aa 395-534) that faces the cytosol. We fused the three cytoplasmic domains of Atg9 to GST, purified them from bacteria and performed in vitro binding experiments with purified Sec23/Sec24 complex. We found that the Sec23/Sec24 complex bound in vitro to GST-Atg9C. Importantly, binding is specific, appears to be saturable and dependent on the presence of Sec24 in the complex (Figure 4). We also showed Sec24 phosphorylation enhances the interaction of the Sec23/Sec24 complex with Atg9 (Figure 5). Therefore, our data implies the *sec24* triple alanine and double alanine mutants are binding deficient mutants (Figure 5).

We also made truncations of the N and C terminal domains of Atg9, as well as portions of the core middle domain, to examine the interaction of Sec24 with Atg9 in vivo. Unfortunately, truncated Atg9 rapidly degraded during the cell lysis procedure used for immunoprecipitation. We encountered similar problems when we expressed fragments of Sec24 in yeast. Nonetheless, the sum total of our findings show that phosphorylation of the conserved membrane distal α helix of Sec24 increases the binding affinity of the Sec23/Sec24 complex with the C-terminus of Atg9 in vitro and in vivo.

*2) The association of the COPII vesicles and Atg9 vesicles should be extensively investigated using biochemical methods and dependency on the Sec24 phosphorylation and the Sec24-Atg9 binding should also be tested.*

We agree that understanding the association of Atg9 vesicles with COPII vesicles is an important question. We have now extensively studied the interaction of Sec24 with domains of Atg9 (described in Point 1 above, also see new data in Figure 4, Figure 5 and the corresponding supplemental data). We initially attempted to form Atg9 vesicles in vitro and monitor their fusion with COPII vesicles. However, the yield of Atg9 vesicles formed in vitro was very low, and the Atg8 lipidation reaction is not very efficient in yeast, leading us to abandon this approach. More recently, in collaboration with Thomas Wollert’s lab, we have been examining the interaction of COPII-coated liposomes with liposomes containing Atg9. Consistent with our in vitro binding data (Figure 4), preliminary experiments using the Atg9 core incorporated into liposomes (aa 281-779 in Figure 4) showed no specific interaction with coated vesicles containing Sec24. These experiments were complicated by the lipid composition used in the Atg9 vesicle preparation, and the affinity of each component for lipids. We are thus continuing to collaborate with Thomas’s lab to better dissect the membrane-related effects of Sec24 and Atg9, but this avenue goes beyond the scope of this manuscript.

*3) Figure 3: Showing autophagy-dependent cleavage of GFP-Atg8 by immunoblotting will complement the results.*

We now include a gel to show the cleavage of GFP-Atg8 in the s*ec24* triple alanine mutant (Figure 2).

*4) It is not clear whether Hrr25 contributes to autophagy indeed through Sec24 phosphorylation. This can be tested by expression of the phosphomimetic Sec24 SE mutant in hrr25-5 mutant cells. The binding between Sec24 SE and Atg9 should be tested in hrr25 mutant cells (Figure 5).*

This was an excellent suggestion. We now show that Sec24 T325E/T328E, but not Sec24 T325A/T328A, partially restores the interaction of Sec24 with Atg9 in the *hrr25-5* mutant (see Figure 6). Furthermore, since demonstrating that Hrr25 contributes to autophagy through Sec24 phosphorylation is a key point in the manuscript, we have also added additional data that strengthen this point. We now show that Sec24 T325E/T328E, but not Sec24 T325A/T328A, largely restores the autophagy defect in the *hrr25-5* mutant (see Figure 6). We demonstrate this in two ways: by analyzing the translocation of GFP-Atg8 to the vacuole (Figure 6), and by examining the cleavage of GFP-Atg8 in the vacuole (Figure 6 and Figure 6—figure supplement 1).

Additionally, we measured the accumulation of autophagic bodies in the *hrr25-5* mutant by transmission electron microscopy (Figure 5—figure supplement 5). This analysis revealed that like the *sec24* triple alanine mutant, the *hrr25-5* mutant accumulates less autophagic bodies than wild- type, indicating that there is a reduction in the number of autophagosomes that form in the absence of Hrr25 kinase activity. Furthermore, as we found with the *sec24* triple alanine mutant, the loss of Hrr25 activity does not affect autophagosome size.

*Reviewer #2*

*My opinion is mixed. The study provides a nice assessment of phosphosites in Sec24 and mutation of the distal threonine residues (324, 328, 328) clearly influence autophagosome abundance. However, I felt the evidence that starvation activates Hrr25 to directly phosphorylate these residues and that phosphorylated Sec24 then reprograms COPII vesicles for targeting to Atg9 containing membranes was a bit preliminary or overstated. At this stage I don't think it is up to par for eLife but I could have missed something.*

*From the consultation session:*

*In reading over your reviews we appear to see similar strengths but share some of the same concerns. I think a current limitation is in clearly placing Sec23-Sec24 activity in autophagosome biogenesis. Phosphorylation of Sec24 appears to play a regulatory role but this could be in somehow re-configuring ERES or in recruiting factors or directing COPII vesicles to a specific target. To devise assays/methods to address this issue is likely to take more than a couple of months which is why my enthusiasm was moderate.*

We agree that placing Sec24 phosphorylation at a specific point in the autophagy cascade is difficult. Therefore, we have used several approaches to address this question. We included new data that suggests Sec24 phosphorylation does not re-configure ERES (Figure 5—figure supplement 3). We also show Sec24 phosphorylation and Hrr25 do not indirectly disrupt autophagy by blocking the assembly of the Atg machinery at the PAS (Figure 5—figure supplement 2; Figure 6—figure supplement 3). Additionally, we now demonstrate that Sec24 phosphorylation directly regulates the interaction of the Sec23/Sec24 complex with Atg9 (Figure 5) and this interaction is needed for autophagy (Figure 6). Our results indicate Sec24 interacts with Atg9 after Atg9 traffics to the PAS (Figure 5—figure supplement 2). Absent from a complete in vitro reconstitution, we believe our efforts to place these regulatory events in context are extensive, and are consistent with the model presented in Figure 7.

*Reviewer #3*

*The membrane supplying autophagosome biogenesis has been difficult to clarify. Recent evidence in yeast and mammals indicate the involvement of COPII vesicles as a contribution to the autophagic membrane. However, how COPII vesicles are diverted from ER-Golgi trafficking to support autophagosome biogenesis was not clear. Some previous evidence shed light on the possible mechanism. For example, the Ge L et al. indicated a spatial regulation by generating the COPII from the ERGIC to separate from the COPII for trafficking which is from the ERES (Ge L et al. eLife 2014) in mammalian cells. Yeast cells lack a special ERGIC. Nonetheless, COPII is still essential for autophagosome biogenesis as shown by the last author's work as well as the Ohsumi group. Since there is no obvious solution to spatially separate the autophagic COPII from the ER-Golgi trafficking COPII in yeast, different mechanisms should be involved. Based on the authors' previous work, they hypothesized that phosphorylation of COPII components should be a reasonable mechanism accounting for the diversion of COPII to autophagy. In the current study, the authors analyzed the phosphorylation of Sec24 because Graef et al. found an association between Sec24 and Atg9 in a previous large scale proteomic analysis (Graef et al. MBoC 2013). They identified the phosphorylation sites of Sec24 and created different loss or gain of function mutations. They found that three membrane distal phosphorylation sites are required for the number of autophagosomes formed during starvation possibly due to their association with Atg9. The Hrr25 is the kinase required for Sec24 phosphorylation and the proposed process shown in the final model. In general, the story is a nice extension of their previous studies. The new advance is to employ the phosphorylation to explain the selection of COPII function. More experiments are necessary to validate the process.*

We would like to clarify that our motivation for studying Sec24 phosphorylation predates the proteomics study that linked multiple COPII coat subunits to Atg9. Since we first reported the COPII coat has targeting functions (Cai et al., 2007 Nature vol445: 941-944), we have been trying to understand the targeting function of the coat. Our interest in understanding the role of coat phosphorylation began when we reported that Sec23 and Sec24 are phosphorylated by Hrr25 (Lord et al., 2011 Nature vol 473:181-186). We began analyzing the role of Sec24 and Sec23 phosphosites in autophagy when we found Hrr25 acts in macroautophagy (Wang et al. 2015 J Cell Biol vol 210:273-285).

Furthermore, we believe the mechanism we describe here is likely to be conserved from yeast to higher eukaryotes. The phosphosites in Sec24 are conserved and Sec24 binds to the C- terminus of Atg9, which is conserved in vertebrates. It is also well known that the functions of Hrr25 are conserved from yeast to man. Our studies differ from the Ge et al. work in that they describe a molecular mechanism by which COPII vesicles recognize the Atg machinery. Even if the COPII vesicles used in autophagosome formation are spatially separated from the vesicles that traffic between the ER and Golgi in mammals, the vesicles used in autophagy have to recognize the Atg machinery. We have added a new section in the Discussion to address how our work relates to Ge et al. 2014.

*1) It is not clear at which stage does the Sec24 phosphorylated COPII act. Although the authors showed no effect on Atg9 on PAS, other marker should also be tested such as Atg14, Atg1, Atg17, Atg2, Atg18 and Atg5. Also, it is not shown if Sec24 mutations, such as the 3A or 3E that affects autophagy, also influence the localization of Sec24 on the PAS.*

As described above in the response to reviewer #2, our data now show that Sec24 (see new data in Figure 5—figure supplement 2) and Hrr25 (see new data in Figure 6—figure supplement 3) are not required for the assembly of the Atg machinery. To show this, we tested a member of each of the groups in the Atg hierarchy (see asterisk in each group in Figure 5—figure supplement 2) and demonstrated they are all recruited normally to the PAS in the *sec24* triple alanine and *hrr25-5* mutants. We also found that the recruitment of Atg14 and Atg2 are normal in the *sec12-4* mutant at its restrictive temperature of 37oC (Figure 6—figure supplement 4). Therefore, our data with the *sec12* mutant are consistent with the data we obtained with the *sec24* triple alanine and *hrr25-5* mutants.

With regard to the second part of this concern (i.e. Also, it is not shown if Sec24 mutations, such as the 3A or 3E that affects autophagy, also influence the localization of Sec24 on the PAS), as we previously reported for *hrr25-5*, we did not see an effect on the localization of COPII vesicles at the PAS with Sec24-3A. But to properly interpret this experiment, we need to do an epistasis experiment in the *sec24Δiss1Δ* strain background with an *atg* mutant that disrupts localization (see Figure Supplement 5 in Wang et al. 2015 J Cell Biol vol 210:273-285). Despite our continued efforts, we have been unable to make this mutant. However, we now show that ERES are not disrupted in the *sec24* triple alanine mutant (Figure 5—figure supplement 3). The ERES were marked by Sec13, not Sec24, for this experiment as it simplified the experiment (i.e. simpler than adding the triple alanine mutations on Sec24-3X-GFP and integrating tagged Sec24 into the genome). ERES in the triple *sec24* phosphomimetic mutant could not be examined as this strain is inviable.

*2) The binding between Sec24 and Atg9 is weak. And the consequence of the interaction is not clear without more biochemical data and interaction site information. The author should find the cytosolic domain of Atg9 that associates with Sec24 and validate with* in vitro *pull down. Based on this notion, single mutation of this Atg9 domain should also achieve a similar effect of abolishment of Sec24 phosphorylation. Or, the author could express a peptide either from Atg9 or Sec24 to compete off the interaction to further confirm.*

As described in detail above to reviewer #1 (see comment #1), we now map the domain in Atg9 that interacts with Sec24.

*3) Although the model of COPII-Atg9 vesicle fusion looks reasonable and has appeared many times in many of their previous papers, it is not supported by any direct evidence.* in vitro *fusion should be performed or the fusion should be analyzed* in vivo *by live cell imaging. The Sec24 mutants should be employed as negative controls.*

We agree that examination of the interaction between COPII vesicles and Atg9 vesicles is a next key question. However, as described above (see comment #2 to Reviewer #1), this is a complex problem to dissect and remains beyond the scope of the current study. Please note that we provide support for the proposal that COPII vesicles fuse with Atg9 vesicles in Figure 7—figure supplement 1 and paragraph #1 in the Discussion. While this experiment does not provide direct proof for our model, it does support our model as only some mutants that disrupt autophagy lead to the accumulation of COPII vesicles at the PAS when autophagy is blocked (also see the last section of the Results entitled “Hrr25 regulates autophagy via Sec24”). Live cell imaging experiments are unlikely to be productive in yeast given their small size and the difficulty in tracking puncta that change composition over time. However, we are currently doing experiments with mammalian cells to support our model and these experiments may be feasible in the future, but are currently beyond the scope of this manuscript.

*From the consultation session:*

*The work is preliminary although has some interesting points. Considering their previous work about the negative effect of SEC23 phosphorylation on autophagy (Wang J et al. JCB 2005) while here it is positive effect of SEC24 phosphorylation on autophagy, the role of COPII phosphorylation and how Hrr25 regulates the specific phosphorylation of SEC23 and SEC24 on different sites and different conditions is really complicated. Although it may take more than 2 months to address these concerns, I believe the authors should be able to address the major concerns of the reviewers to improve the current work. In addition, the mechanism of how Hrr25 selects SEC23 and SEC24 as well as specific phosphorylation sites on SEC24 that specifically affect autophagy pathway and ATG9 association is very important and will increase the impact of the study.*

We agree that understanding how Hrr25 specifically interacts with Sec23 and Sec24 is a key question. We now use complementation experiments to provide additional evidence that Hrr25 acts in autophagy via phosphorylation of the Sec24 membrane distal regulatory surface (Figure 6). These results clarify our earlier finding that Hrr25 is a positive regulator of autophagy (Wang et al. 2015 J Cell Biol vol 210:273-28).

We previously reported that phosphomimetic (but not alanine) mutations at two sites (S742 and T747) in Sec23 disrupt ER-Golgi traffic and autophagy (Lord et al., 2011 Nature vol 473:181-186; Wang et al. 2015 J Cell Biol vol 210:273-28). These phosphomimetic mutations were shown to interfere with the binding of TRAPP (Ypt1/Rab1 GEF) to Sec23 (Lord et al., 2011 Nature vol 473:181-186). Technically, these previous data showed that dephosphorylation (not phosphorylation) of Sec23 is required for both autophagy and ER-Golgi traffic.

[Editors' note: the author responses to the re-review follow.]

[…]

*Essential revisions:*

*1) Figure 3. It is hard to tell that the GFP-Atg8 structures indeed represent autophagosomes rather than precursor structures (phagophores) and the PAS. The size of phagophores and the PAS are smaller than that of closed autophagosomes. It would be ideal to measure the size of GFP-Atg8 structures in ypt7Δ cells, in which closed autophagosomes should accumulate.*

In response to this point, I wrote to the editorial office stating “we believe we already addressed concern #1 by the EM analysis we performed (Figure 3).” We received a positive response (below) that we have addressed by changes to the figures and text. We hope this clarifies our conclusions.

Response from editors:

"We understand that the EM data convincingly show the decrease in the number not size of autophagic bodies in sec24 triple alanine mutant cells. However, the fluorescence data do not support this conclusion as explained in the previous comments. If the authors wish to retain the fluorescence data, it is important to state that the GFP-Atg8 structures represent either the PAS, phagophore, or autophagosome, and it is difficult to conclude that the size of autophagosome is not affected. For example, the Y-axis of Figure 3 cannot be "autophagosome number/diameter". These should be "GFP-Atg8 puncta number/diameter (/cell)".

We believe the SIM data contributes to our study, so we have retained the data, but we have changed the Y-axis in Figure 3 and Figure 3—figure supplement 1 to "GFP-Atg8 puncta number/diameter (/cell)". The text and legends have been changed accordingly. We now make it clear that the GFP-Atg8 puncta could represent either the PAS, phagophores or autophagosomes.

*2) Related to the above criticism, it is also difficult to distinguish small autophagosomes from the PAS under nutrient-rich conditions. The presence of small GFP-Atg8 puncta does not necessarily mean that Cvt vesicles are normally formed because they may merely represent the PAS. Thus, the authors' proposal that COPII vesicles contribute to autophagosome formation under starvation but not nutrient-rich conditions (Cvt vesicle formation) should be carefully reevaluated. In fact, it was reported that COPII mutants are defective in the Cvt pathway (Reggiori et al. Mol Biol Cell 15, 2189, 2004). The Cvt pathway should be directly examined by seeing the maturation of Ape1 in the 3A mutant.*

We have directly examined the Cvt pathway, by examining Ape1 maturation, and now show there is no processing defect in the *sec24* alanine mutants (see Figure 3—figure supplement 1). In the isogenic wild-type strain we use, Ape1 is partially processed. Nonetheless, it is clear that the *sec24* alanine mutants process Ape1 as well as wild-type. We also note that, while Reggioro et al. 2004 (MBoC 15, 2189) concluded that COPII vesicles are required on the Cvt pathway, Ishihara et al. 2001 (MBoC 12, 3690) disagree. Their EM analysis showed Cvt vesicle formation occurs normally in a *sec12* mutant. This is consistent with our observations.

*3) It is difficult to conclude that COPII vesicles localize to the PAS by simply observing Sec13 puncta, most of which likely represent ER exit sites. It would be difficult to distinguish the ERESs and COPII vesicles by fluorescence microscopy. Other groups concluded that the PAS forms in the vicinity of the ER exit site based on results of similar experiments.*

This is a good point and we should have been more careful with our phrasing. We now refer to Sec13 puncta as COPII coated structures, which includes ERES and COPII vesicles (see first paragraph in the Discussion).

*4) The title and/or Abstract should provide a clear indication of the biological system under investigation (i.e., species name or broader taxonomic group, if appropriate). Please revise your title and/or abstract with this advice in mind.*

The species is now mentioned in the Abstract.